# Towards a Better Global Loss Landscape of GANs

**Ruoyu Sun**[*], **Tiantian Fang,  Alex Schwing**
University of Illinois at Urbana-Champaign
`ruoyus,tf6,aschwing@illinois.edu`

## Abstract

Understanding of GAN training is still very limited. One major challenge is its non-convex-non-concave min-max objective, which may lead to sub-optimal local minima. In this work, we perform a global landscape analysis of the empirical loss of GANs. We prove that a class of separable-GAN, including the original JS-GAN, has exponentially many bad basins which are perceived as mode-collapse. We also study the relativistic pairing GAN (RpGAN) loss which couples the generated samples and the true samples. We prove that RpGAN has no bad basins. Experiments on synthetic data show that the predicted bad basin can indeed appear in training. We also perform experiments to support our theory that RpGAN has a better landscape than separable-GAN. For instance, we empirically show that RpGAN performs better than separable-GAN with relatively narrow neural nets. The code is available at `https://github.com/AilsaF/RS-GAN`.

## 1   Introduction

Generative Adversarial Nets (GANs) [35] are a successful method for learning data distributions. Current theoretical efforts to advance understanding of GANs often focus on statistics or optimization.

On the statistics side, Goodfellow et al. [35] built a link between the min-max formulation and the J-S (Jenson-Shannon) distance. Arjovsky and Bottou [3] and Arjovsky et al. [4] proposed an alternative loss function based on the Wasserstein distance. Arora et al. [5] studied the generalization error and showed that both the Wasserstein distance and J-S distance are not generalizable (i.e., both require an exponential number of samples). Nevertheless, Arora et al. [5] argue that the real metric used in practice differs from the two statistical distances, and can be generalizable with a proper discriminator. Bai et al. [7] and Lin et al. [58] analyzed the potential "lack of diversity": two different distributions can have the same loss, which may cause mode collapse. Bai et al. [7] argue that proper balancing of generator and discriminator permits both generalization and diversity.

On the optimization side, cyclic behavior (non-convergence) is well recognized [65, 8, 34, 11]. This is a generic issue for min-max optimization: a first-order algorithm may cycle around a stable point, converge very slowly or even diverge. The convergence issue can be alleviated by more advanced optimization algorithms such as optimism (Daskalakis et al. [23]), averaging (Yazıcı et al. [88]) and extrapolation (Gidel et al. [33]).

Besides convergence, another general optimization challenge is to avoid sub-optimal local minima. It is an important issue in non-convex optimization (e.g., Zhang et al. [91], Sun [81]), and has received great attention in matrix factorization [31, 14, 19] and supervised learning [38, 47, 2, 92, 27]. For GANs, the aforementioned works [65, 8, 34, 11] either analyze convex-concave games or perform local analysis. Hence they do not touch the global optimization issue of non-convex problems. Mescheder et al. [65] and Feizi et al. [30] prove global convergence only for simple settings where the true data distribution is a single point or a single Gaussian distribution. The global analysis of GANs for a fairly general data distribution is still a rarely touched direction.

---

[*]Corresponding author

Table 1: Comparison of theoretical works.

| | Supervised Learning | | GANs | |
|---|---|---|---|---|
| | paper | brief description | paper | brief description |
| Generalization analysis | [9] | generalization bound for neural-nets | [5] | generalization bound for GANs |
| Convergence analysis | [77] | convex problem, divergence of Adam convergence of AMSGrad | [23] | bi-linear game, non-convergence of GDA convergence of optimistic GDA |
| Global landscape | [73] [50] | Any distinct input data Wide neural-nets have no sub-optimal basins | **This work** | Any distinct input data SepGAN has bad basins; RpGAN does not |

* This table does NOT show a complete list of works. The goal is to list various types of works. Only one or two works are listed as examples of that class. Results on global convergence (e.g. [38, 2, 27]) for supervised learning are not listed in the table, because there are no similar results for GANs yet.

The global analysis of GANs is an interesting direction for the following reasons. **First**, from a theoretical perspective, it is an indispensable piece for a complete theory. To put our work in perspective, we compare representative works in supervised learning with works on GANs in Tab. 1. **Second**, it may help to understand mode collapse. Bai et al. [7] conjectured that a lack of diversity may be caused by optimization issues, albeit convergence analysis works [65, 8, 34, 11] do not link non-convergence to mode collapse. Thus we suspect that mode collapse is at least partially related to sub-optimal local minima, but a formal theory is still lacking. **Third**, it may help to understand the training process of GANs. Even understanding a simple two-cluster experiment is challenging because the loss values of min-max optimization are fluctuating during training. Global analysis can provide an additional lens in demystifying the training process.

Additional related work is reviewed in Appendix A.

**Challenges and our solutions.** While the idea of a global analysis is natural, there are a few obstacles. First, it is hard to follow a common path of supervised learning [38, 47, 2, 92, 27] to prove global convergence of gradient descent for GANs, because the dynamics of non-convex-non-concave games are much more complicated. Therefore, we resort to a *landscape analysis*. Note that our approach resembles an "equilibrium analysis" in game theory. Second, it was not clear which formulation can cure the landscape issue of JS-GAN. Wasserstein GAN (W-GAN) is a candidate, but its landscape is hard to analyze due to the extra constraints. After analyzing the issue of JS-GAN, we realize that the idea of "paring" (pair the true data and generated data), which is implicitly used by W-GAN, may cure the issue. However, W-GAN is a constrained formulation which seems hard to analyze, thus we consider a formulation that has the "pairing" component but is unconstrained: relativistic pairing GANs (RpGANs) [41, 42] [2]. We prove that RpGANs have a better landscape than separable-GANs (generalization of JS-GAN). Third, it was not clear whether the theoretical finding affects practical training. We make a few conjectures based on our landscape theory and design experiments to verify those. Interestingly, the experiments match the conjectures quite well.

**Our contributions.** This work provides a global landscape analysis of the empirical version of GANs. Our contributions are summarized as follows:

- *Does the original JS-GAN have a good landscape, provably?* For JS-GAN [35], we prove that the outer-minimization problem has exponentially many sub-optimal strict local minima. Each strict local minimum corresponds to a mode-collapse situation. We also extend this result to a class of separable-GANs, covering hinge loss and least squares loss.
- *Is there a way to improve the landscape, provably?* We study a class of relativistic paring GANs (RpGANs) [41] that pair the true data and the generated data in the loss function. We prove that the outer-minimization problem of RpGAN has no bad strict local minima, improving upon separable-GANs.
- *Does the improved landscape lead to any empirical benefit?* Based on our theory, we predict that RpGANs are more robust to data, network width and initialization than their separable counter-parts, and our experiments support our prediction. Although the empirical benefit of RpGANs was observed before [41], the aspects we demonstrate are closely related to our landscape theory. In addition, using synthetic experiments we explain why mode-collapse (as bad basins) can slow down JS-GAN training.

## 2  Difference of Population Loss and Empirical Loss

Goodfellow et al. [35] proved that the population loss of GANs is convex in the space of probability densities. We highlight that this convexity highly depends on a simple property of the population loss, which may vanish in an empirical setting.

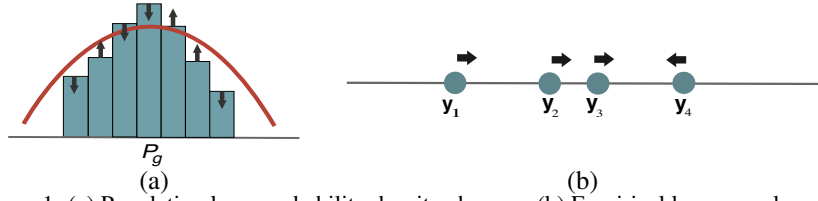

(a)            (b)

Figure 1: (a) Population loss: probability density changes; (b) Empirical loss: samples move.

Suppose $p_{\text{data}}$ is the data distribution, $p_{\text{g}}$ is a generated distribution and $D \in C_{(0,1)}(\mathbb{R}^d)$, where $C_{(0,1)}(\mathbb{R}^d)$ is the set of continuous functions with domain $\mathbb{R}^d$ and codomain $(0,1)$. Consider the JS-GAN formulation [35]

$$\min_{p_{\text{g}}} \phi_{\text{JS}}(p_{\text{g}}; p_{\text{data}}), \text{ where } \phi_{\text{JS}}(p_{\text{g}}; p_{\text{data}}) = \sup_{D} \mathbb{E}_{x \sim p_{\text{data}}, y \sim p_{\text{g}}}[\log(D(x)) + \log(1 - D(y))].$$

**Claim 2.1.** *([35, in proof of Prop. 2]) The objective function $\phi_{\text{JS}}(p_{\text{g}}; p_{\text{data}})$ is convex in $p_{\text{g}}$.*

The proof utilizes two facts: first, the supremum of (infinitely many) convex functions is convex; second, $\mathbb{E}_{x \sim p_{\text{data}}, y \sim p_{\text{g}}}[\log(D(x)) + \log(1 - D(y))]$ is a linear function of $p_{\text{g}}$. The second fact is the essence of the argument, which we restate below in a more general form.

**Claim 2.2.** $\mathbb{E}_{y \sim p_{\text{g}}}[f^{\text{arb}}(y)]$ *is a linear function of $p_{\text{g}}$, where $f^{\text{arb}}(y)$ is an arbitrary function of $y$.*

Claim 2.2 implies that $\min_{p_{\text{g}}} \mathbb{E}_{y \sim p_{\text{g}}}[f^{\text{arb}}(y)]$ is a convex problem. One approach to solve it is to draw finitely many samples (particles) $y_i, i = 1, \ldots, n$ from $p_{\text{g}}$, and approximate the population loss by the empirical loss. See Fig. 1 for a comparison of the probability space and the particle space. For an arbitrarily complicated function such as $f^{\text{arb}}(y) = \sin(\|y\|^8 + 2\|y\|^3 + \log(\|y\|^4 + 1))$, the population loss is convex in $p_{\text{g}}$, but clearly the empirical loss is non-convex in $(y_1, \ldots, y_n)$. This example indicates that studying the empirical loss may better reveal the difficulty of the problem (especially with a limited number of samples). See Appendix G for more discussions.

We focus on the empirical loss in this work. Suppose there are $n$ data points $x_1, \ldots, x_n$. We sample $n$ latent variables $z_1, \ldots, z_n \in \mathbb{R}^{d_z}$ according to a rule (e.g., i.i.d. Gaussian) and generate artificial data $y_i = G(z_i), i = 1, \ldots, n$. The empirical version of JS-GAN addresses $\min_Y \phi_{\text{JS}}(Y, X)$ where

$$\phi_{\text{JS}}(Y, X) \triangleq \sup_{D} \frac{1}{2n} \sum_{i=1}^{n} [\log(D(x_i)) + \log(1 - D(y_i))]. \tag{1}$$

Note that the empirical loss is considered in Arora et al. [5] as well, but they study the generalization properties. We focus on the optimization properties, which is complementary to their work.

## 3 Landscape Analysis of GANs: Intuition and Toy Results

In this section, we discuss the main intuition and present results for a 2-point distribution.

**Intuition of Bad "Local Minima" and Separable-GAN:** Consider an empirical data distribution consisting of two samples $x_1, x_2 \in \mathbb{R}$. The generator produces two data points $y_1, y_2$ to match $x_1, x_2$. We illustrate the training process of JS-GAN in Fig. 2. Initially, $y_1, y_2$ are far from $x_1, x_2$, thus the discriminator can easily separate true data and fake data. After the generator update, $y_1, y_2$ cross the decision boundary to fool the discriminator. Then, after the discriminator update, the decision boundary moves and can again separate true data and fake data. As iterations progress, $y_1, y_2$ and the decision boundary may stay close to $x_1$, causing mode-collapse.

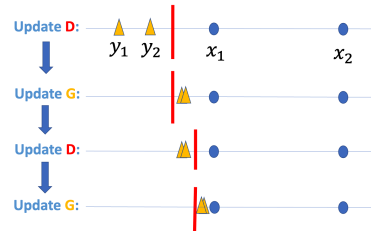

Figure 2: Issue of separable-GAN (including JS-GAN). After updating $G$, fake data crosses boundary to fool $D$; after updating $D$, they are separated by $D$. Fake data may be stuck near $x_1$.

The intuition above is the starting point of this work. We notice that Unterthiner et al. [83], Li and Malik [53] presented somewhat similar intuition, and Kodali et al. [45] suggested the connection between mode collapse and a bad equilibrium. Nevertheless, Li and Malik [53], Kodali et al. [45] do not present a theoretical result, and Unterthiner et al. [83] uses a significantly different formulation from standard GANs. See Appendix A for more.

We point out that a major reason for the above issue is a single decision boundary which judges the generated samples. Therefore, this issue exists not only for the JS-GAN, but also for a large class of GANs which we call separable-GANs:

$$\min_Y \sup_f \sum_{i=1}^n h_1(f(x_i)) + h_2(-f(y_i)), \tag{2}$$

where $h_1, h_2$ are fixed scalar functions, such as $h_1(t) = h_2(t) = -\log(1 + e^{-t})$ and $h_1(t) = h_2(t) = -\max\{0, 1-t\}$, and $f$ is chosen from a function space (e.g., a set of neural-net functions).

**Pairing as Solution: Rp-GAN.** A natural solution is to use a different "decision boundary" for every generated point, e.g., pairing $x_i$ and $y_i$, as illustrated in Fig. 3.

A suitable loss is the relativistic paring GAN (RpGAN)[3]

$$\min_Y \sup_f \sum_{i=1}^n h(f(x_i) - f(y_i)), \tag{3}$$

where $h$ is a fixed scalar function and $f$ is chosen from a function space. RS-GAN (relative standard GAN) is a special case where $h(t) = -\log(1 + e^{-t})$. More specifically, RS-GAN addresses $\min_Y \phi_{\mathrm{RS}}(Y, X)$ where

Figure 3: Idea of RpGAN: breaking locality by "personalized" judgement.

$$\phi_{\mathrm{RS}}(Y, X) \triangleq \sup_f \frac{1}{n} \sum_{i=1}^n \log \frac{1}{1 + \exp(f(y_i) - f(x_i)))}. \tag{4}$$

W-GAN [3] can be viewed as a variant of RpGAN where $h(t) = t$, with extra Lipschitz constraint.

We wonder how the issue of seperable-GANs relates to "local minima" and how "pairing" helps. We present results for JS-GAN and RS-GAN for the two-point case below.

**Global Landscape of 2-Point Case:** Depending on the positions of $y_1, y_2$, there are four states $s_0, s_{1a}, s_{1b}, s_2$. They represent the four cases $|\{x_1, x_2\} \cap \{y_1, y_2\}| = 0$, $y_1 = y_2 \in \{x_1, x_2\}$, $|\{x_1, x_2\} \cap \{y_1, y_2\}| = 1$, and $\{x_1, x_2\} = \{y_1, y_2\}$ respectively. Training often starts from the "no-recovery" state $s_0$, and ideally should end at the "perfect-recovery" state $s_2$. There are two intermediate states: $s_{1a}$ means all generated points fall into one mode ("mode collapse"); $s_{1b}$ means one generated point is the true data point while the other is not a desired data point, which we call "mode dropping"[4]. The first three states can transit to each other (assuming continuous change of $Y$), but only $s_{1b}$ can transit to $s_2$. We illustrate the landscape of $\phi_{\mathrm{JS}}(Y; X)$ and $\phi_{\mathrm{RS}}(Y; X)$ in Fig. 4, by indicating the values in different states. The detailed computation is given next.

**JS-GAN 2-Point Case:** The range of $\phi_{\mathrm{JS}}(Y, X)$ is $[-\log 2, 0]$. The value for the four states are:

**Claim 3.1.** *The minimal value of $\phi_{\mathrm{JS}}(Y, X)$ is $-\log 2$, achieved at $\{y_1, y_2\} = \{x_1, x_2\}$.*

$$\phi_{\mathrm{JS}}(Y, X) = \begin{cases} -\log 2 \approx -0.6931 & \text{if } \{x_1, x_2\} = \{y_1, y_2\}, \\ -\log 2/2 \approx -0.3467 & \text{if } |\{x_1, x_2\} \cap \{y_1, y_2\}| = 1, \\ \frac{1}{4}(2\log 2 - 3\log 3) \approx -0.4774 & \text{if } y_1 = y_2 \in \{x_1, x_2\}, \\ 0 & \text{if } |\{x_1, x_2\} \cap \{y_1, y_2\}| = \emptyset. \end{cases}$$

We illustrate the landscape of $\phi_{\mathrm{JS}}(Y, X)$ in Fig. 4(a). As a corollary of the above claim, the outer optimization of the original GAN has a bad strict local minimum at state $s_{1a}$ (a mode-collapse).

**Corollary 3.1.** $\bar{Y} = (x_1, x_1)$ *is a sub-optimal strict local-min of the function $g(Y) = \phi_{\mathrm{JS}}(Y, X)$.*

**RS-GAN 2-Point Case:** The range is still $\phi_{\mathrm{RS}}(Y, X) \in [-\log 2, 0]$. The values are:

**Claim 3.2.** *The minimal value of $\phi_{\mathrm{RS}}(Y, X)$ is $-\log 2$, achieved at $\{y_1, y_2\} = \{x_1, x_2\}$. In addition,*

$$\phi_{\mathrm{RS}}(Y, X) = \begin{cases} -\log 2 \approx -0.6931 & \text{if } \{x_1, x_2\} = \{y_1, y_2\}, \\ -\frac{1}{2}\log 2 \approx -0.3466 & \text{if } |\{i : \exists j, \text{ s.t. } x_i = y_j\}|, \\ 0 & \text{otherwise.} \end{cases}$$

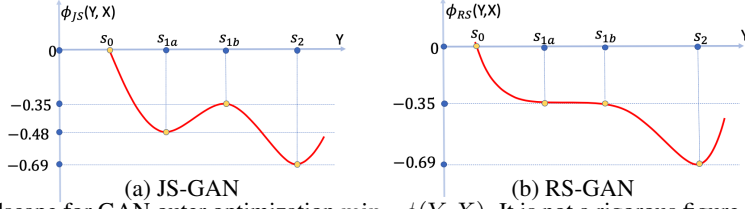

Figure 4: Landscape for GAN outer optimization $\min_Y \phi(Y, X)$. It is not a rigorous figure because: (i) there are only four possible values, thus the function is piece-wise linear while we use smooth curves for accessibility. (ii) the landscape should be two-dimensional, but we illustrate them in 1D space. Nevertheless, it is still useful for understanding GAN training, as discussed later in Section 5 and Appendix B.

We illustrate $\phi_{\mathrm{RS}}(Y, X)$ in Fig. 4(b). Importantly, note that the only basin is the global minimum. In contrast, the landscape of JS-GAN contains a bad basin at a mode-collapsed pattern.

The proofs of Claim 3.1 and Claim 3.2 are given in Appendix H. We briefly explain the main insight provided by these proofs. For the mode-collapsed pattern $s_{1\mathrm{a}}$, the loss value of JS-GAN is $-\frac{1}{4}\min_{s,t}[\log(1+e^{-t})+2\log(1+e^t)+\log(1+e^{-s})] = \frac{1}{4}(\log\frac{1}{3}+2\log\frac{2}{3}) \approx -0.48 \neq -\frac{r}{2}\log 2$ for any integer $r$. This creates an "irregular" value among other loss values of the form $-\frac{r}{2}\log 2$. In contrast, for pattern $s_{1\mathrm{a}}$, the loss value of RS-GAN is $-\frac{1}{2}\min_{s,t}[\log(1+e^{t-t})+\log(1+e^{t-s})] = -\frac{1}{2}\log 2$, which is of the form $-\frac{r}{2}\log 2$. Therefore, for the 2-point case, RS-GAN has a better landscape because it avoids the "irregular" value of JS-GAN due to its "pairing". This insight is the foundation of the general theory presented in the next section.

# 4 Main Theoretical Results

## 4.1 Landscape Results in Function Space

We present our main theoretical results, extending the landscape results from $n = 2$ to general $n$.

Denote $\xi(m) \triangleq \sup_{t \in \mathbb{R}}(h_1(t) + mh_2(-t))$.

**Assumption 4.1.** $\sup_{t \in \mathbb{R}} h_1(t) = \sup_{t \in \mathbb{R}} h_2(t) = 0$.

**Assumption 4.2.** $\xi(m) > m\xi(1)$, $\forall\, m \in [2, n]$.

**Assumption 4.3.** $\xi(m) < \xi(m-1)$, $\forall\, m \in [1, n]$.

It is easy to prove that under Assumption 4.1, $\xi(m-1) \geq \xi(m) \geq m\xi(1)$ always holds. Assumption 4.2 and Assumption 4.3 require strict inequalities, thus do not always hold (e.g., for constant functions). Nevertheless, most non-constant functions satisfy these assumptions.

The separable-GAN (SepGAN) problem (empirical loss, function space) is

$$\min_{Y \in \mathbb{R}^{d \times n}} g_{\mathrm{SP}}(Y), \text{ where } g_{\mathrm{SP}}(Y) = \frac{1}{2n} \sup_{f \in C(\mathbb{R}^d)} \sum_{i=1}^{n} [h_1(f(x_i)) + h_2(-f(y_i))]. \tag{5}$$

**Theorem 1.** *Suppose $x_1, x_2, \ldots, x_n \in \mathbb{R}^d$ are distinct. Suppose $h_1, h_2$ satisfy Assumptions 4.1, 4.2 and 4.3. Then for separable-GAN loss $g_{\mathrm{SP}}(Y)$ defined in Eq. (5), we have: (i) The global minimal value is $-\frac{1}{2}\sup_{t\in\mathbb{R}}(h_1(t)+h_2(-t))$, which is achieved iff $\{y_1, \ldots, y_n\} = \{x_1, \ldots, x_n\}$. (ii) If $y_i \in \{x_1, \ldots, x_n\}, i \in \{1, 2, \ldots, n\}$ and $y_i = y_j$ for some $i \neq j$, then $Y$ is a sub-optimal strict local minimum. Therefore, $g_{\mathrm{SP}}(Y)$ has $(n^n - n!)$ sub-optimal strict local minima.*

Remark 1: $h_1(t) = h_2(t) = -\log(1 + e^{-t})$ satisfy Assumptions 4.1, 4.2 and 4.3, thus Theorem 1 applies to JS-GAN. It also applies to hinge-GAN with $h_1(t) = h_2(t) = -\max\{0, 1-t\}$ and LS-GAN (least-square GAN) with $h_1(t) = -(1-t)^2, h_2(t) = -t^2$.

Next we consider RpGANs. The RpGAN problem (empirical loss, function space) is

$$\min_{Y \in \mathbb{R}^{d \times n}} g_{\mathrm{R}}(Y), \text{ where } g_{\mathrm{R}}(Y) = \frac{1}{n} \sup_{f \in C(\mathbb{R}^d)} \sum_{i=1}^{n} [h(f(x_i) - f(y_i))]. \tag{6}$$

**Definition 4.1.** *(global-min-reachable) We say a point $w$ is global-min-reachable for a function $F(w)$ if there exists a continuous path from $w$ to one global minimum of $F$ along which the value of $F(w)$ is non-increasing.*

**Assumption 4.4.** $\sup_{t \in \mathbb{R}} h(t) = 0$ *and* $h(0) < 0$.

**Assumption 4.5.** $h$ *is a concave function in* $\mathbb{R}$.

**Theorem 2.** *Suppose* $x_1, x_2, \ldots, x_n \in \mathbb{R}^d$ *are distinct. Suppose* $h$ *satisfies Assumptions 4.4 and 4.5. Then for RpGAN loss* $g_R$ *defined in Eq. (6): (i) The global minimal value is* $h(0)$*, which is achieved iff* $\{y_1, \ldots, y_n\} = \{x_1, \ldots, x_n\}$*. (ii) Any* $Y$ *is global-min-reachable for the function* $g_R(Y)$*.*

This result sanity checks the loss $g_R(Y)$: its global minimizer is indeed the desired empirical distribution. In addition, it establishes a significantly different optimization landscape for RpGAN.

Remark 1: $h(t) = -\log(1 + e^{-t})$ satisfies Assumption 4.4 and 4.5, thus Theorem 2 applies to RS-GAN. It also applies to Rp-hinge-GAN with $h(t) = -\max\{0, a - t\}$ and Rp-LS-GAN with $h(t) = -(a - t)^2$, for any positive constant $a$.

Remark 2: The W-GAN loss is $\frac{1}{n} \sup_f \sum_i h(f(x_i) - f(y_i))$ where $h(t) = t$; however, since $\sup_t h(t) = \infty$ it does not satisfy Assumption 4.4. The unboundedness of $h(t) = t$ necessitates extra constraints, which make the landscape analysis of W-GAN challenging; see Appendix L. Analyzing the landscape of W-GAN is an interesting future work.

To prove Theorem 1, careful computation suffices; see Appendix I. The proof of Theorem 2 is a bit involved. We first build a graph with nodes representing $x_i$'s and $y_i$'s, then decompose the graph into cycles and trees, and finally compute the loss value by grouping the terms according to cycles and trees and calculate the contribution of each cycle and tree. The detailed proof is given in Appendix J.

## 4.2 Landscape Results in Parameter Space

We now consider a deep net generator $G_w$ with $w \in \mathbb{R}^K$ and a deep net discriminator $f_\theta$ with $\theta \in \mathbb{R}^J$. Different from before, where we optimize over $y_i$ and $f$ (function space), we now optimize over $w$ and $\theta$ (parameter space).

We first present a technical assumption. For $Z = (z_1, \ldots, z_n) \in \mathbb{R}^{d_z \times n}$, $Y = (y_1, \ldots, y_n) \in \mathbb{R}^{d \times n}$ and $\mathcal{W} \subseteq \mathbb{R}^K$, define a set $G^{-1}(Y; Z, \mathcal{W}) \triangleq \{w \in \mathcal{W} \mid G_w(z_i) = y_i, \forall i\}$.

**Assumption 4.6.** *(path-keeping property of generator net): For any distinct* $z_1, \ldots, z_n \in \mathbb{R}^{d_z}$*, any continuous path* $Y(t), t \in [0, 1]$ *in the space* $\mathbb{R}^{d \times n}$ *and any* $w_0 \in G^{-1}(Y(0); Z, \mathcal{W})$*, there is continuous path* $w(t), t \in [0, 1]$ *such that* $w(0) = w_0$ *and* $Y(t) = G_{w(t)}(Z), t \in [0, 1]$*.*

Intuitively, this assumption relates the paths in the function space to the paths in the parameter space, thus the results in function space can be transferred to the results in parameter space. The formal results involve two extra assumptions on representation power of $f_\theta$ and $G_w$ (see Appendix K for details). Informal results are as follows:

**Proposition 1.** *(informal) Consider the separable-GAN problem* $\min_{w \in \mathbb{R}^K} \varphi_{\text{sep}}(w)$*, where*

$$\varphi_{\text{sep}}(w) = \sup_\theta \frac{1}{2n} \sum_{i=1}^{n} [h_1(f_\theta(x_i)) + h_2(-f_\theta(G_w(z_i)))]. \tag{7}$$

*Suppose* $h_1, h_2$ *satisfy the assumptions of Theorem 1. Suppose* $G_w$ *satisfies Assumption 4.6 (with certain* $\mathcal{W}$*). Suppose* $f_\theta$ *and* $G_w$ *have enough representation power (formalized in Appendix K). Then there exist at least* $(n^n - n!)$ *distinct* $w \in \mathcal{W}$ *that are not global-min-reachable for* $\varphi_{\text{sep}}(w)$*.*

**Proposition 2.** *(informal) Consider the RpGAN problem* $\min_{w \in \mathbb{R}^K} \varphi_R(w)$*, where*

$$\varphi_R(w) = \sup_\theta \frac{1}{n} \sum_{i=1}^{n} [h(f_\theta(x_i) - f_\theta(G_w(z_i)))]. \tag{8}$$

*Suppose* $h$ *satisfies the assumptions of Theorem 2. Suppose* $G_w$ *and* $f_\theta$ *satisfy the same assumptions as Proposition 1. Then any* $w \in \mathcal{W}$ *is global-min-reachable for* $\varphi_R(w)$*.*

Remark 1: The existence of a decreasing path does not necessarily mean an algorithm can follow it. Nevertheless, our results already distinguish SepGAN and RpGAN. We will illustrate that these results help demystify GAN training in Sec. 5, and present supporting experiments in Sec. 6.

Remark 2: The two results rely on a few assumptions of neural-nets including Assumption 4.6. These assumptions can be satisfied by certain over-parameterized neural-nets, in which case $\mathcal{W}$ is a certain dense subset of $\mathbb{R}^K$ or $\mathbb{R}^K$ itself. For details see Appendix K.1.

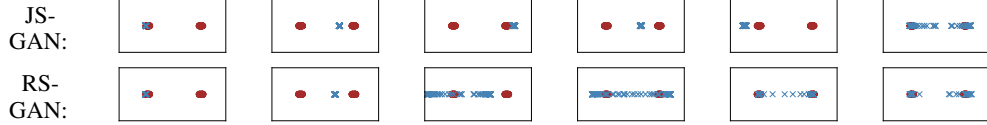

Figure 5: Training process of JS-GAN and RS-GAN for two-cluster data. True data are red, fake data are blue. RS-GAN escapes from mode collapse faster than JS-GAN.

### 4.3 Discussion of Implications

These results distinguish the SepGAN and RpGAN landscapes. Theoretically, there is evidence regarding the benefit of losses without sub-optimal basins. Bovier et al. [17] proved that it takes the Langevin diffusion at least $e^{\omega(h)}$ time to escape a depth-$h$ basin. A recent work [91] proved that the hitting time of SGLD (stochastic gradient Langevin dynamics) is positively related to the height of the barrier, and SGLD may escape basins with low barriers relatively fast. The theoretical insight is that a landscape without a bad basin permits better quality solutions or a faster convergence to good-quality solutions.

We now discuss the possible gap between our theory and practice. We proved that a mode collapse $Y^*$ is a bad basin in the generator space, which indicates that $(Y^*, D^*(Y^*))$ is an attractor in the joint space of $(Y, D)$ and hard to escape by gradient descent ascent (GDA). In GAN training, the dynamics are not the same as GDA dynamics due to various reasons (e.g., sampling, unequal $D$ and $G$ updates), and basins could be escaped with enough training time (e.g., [91]). In addition, a randomly initialized $(Y, D)$ might be far away from the basins at $(Y^*, D^*(Y^*))$, and properly chosen hyper-parameters (e.g., learning rate) may re-position the dynamics so as to avoid attraction to bad basins. Further, it is known that adding neurons can smooth the landscape of deep nets (e.g., eliminating bad basins in neural-nets [50]), thus wide nets might help escape basins in the $(Y, D)$-space faster. In short, the effect of bad basins may be mitigated via the following factors: (i) proper initial $D$ and $Y$; (ii) long enough training time; (iii) wide neural-nets; (iv) enough hyper-parameter tuning. These factors make it relatively hard to detect the existence of bad basins and their influences. We support our landscape theory, by identifying differences of SepGAN and RpGAN in synthetic and real-data experiments.

## 5 Case Study of Two-Cluster Experiments

Although in Section 3 we argue that, *intuitively*, mode collapse can happen for training JS-GAN for two-point generation, it does not necessarily mean mode collapse really appears in practical training. We discuss a two-cluster experiment, an extension of two-point generation, in order to build a link between theory and practice. We aim to understand the following question: does mode collapse really appear as a "basin", and how does it affect training?

Suppose the true data are two clusters around $c_1 = 0$ and $c_2 = 4$. We sample 100 points from the two clusters as $x_i$'s, and sample $z_1, \ldots, z_{100}$ uniformly from an interval. We use 4-layer neural-nets for the discriminator and generator. We use the non-saturating versions of JS-GAN and RS-GAN.

**Mode collapse as bad basin can appear.** We visualize the movement of fake data in Fig. 5, and plot the loss value of D (i.e. discriminator) over iterations in Fig. 6(a,b). Interestingly, the minimal $D$ losses are around $0.48$, which is the value of $\phi_{\mathrm{JS}}$ at state $s_{1a}$. It is easy to check that the optimal $D = D^*(s_{1a})$ for a mode collapse state $s_{1a}$ satisfies $\{D(c_1), D(c_2)\} = \{1, 1/3\}$, and Fig. 6(c) shows that at iteration 2800 the $D$ actually becomes $D^*$. This provides a concrete example that training gets stuck at a mode collapse due to the bad-basin-effect. We also notice that there are a few more attempts to approach the bad attractor $(s_{1a}, D^*(s_{1a}))$ (e.g., from iteration 2000 to 2500). In RS-GAN training, the minimal loss is around $0.35$, which is also the value of $\phi_{\mathrm{RS}}$ at state $s_{1a}$. The attracting power of $(s_{1a}, D^*(s_{1a}))$ is weaker than for JS-GAN, thus only attracts the iterates for a very short time. RS-GAN needs 800 iterations to escape, which is about 3 times faster than the escape for JS-GAN.

**Effect of width:** We see a clear effect of width on convergence speed. As the networks become wider, both JS-GAN and RS-GAN converge faster. We find that the reason of faster convergence is because wider nets make JS-GAN escape mode collapse faster. See details in Appendix B.

More experiment details and findings are presented in Appendix B.

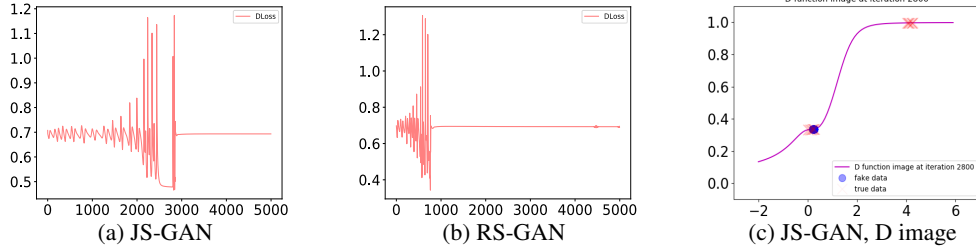

| | (a) JS-GAN | (b) RS-GAN | (c) JS-GAN, D image |

Figure 6: (a) and (b): Evolution of $D$ loss over iterations. RS-GAN is 3-4$\times$ faster than JS-GAN. (c) For JS-GAN training in (a), we plot $(Y, D)$ together at iteration 2800. $Y$ are represented in blue points, and they are near $c_1 = 0$. $D$ is near the optimal $D^*(s_{1a})$ since $D(0) \approx 1/3$ and $D(4) \approx 1$. Interestingly, this bad attractor $(Y, D)$ is similar to the one discussed in Fig. 1, so the intuition of "local-min" is verified in (c).

| | **CIFAR-10** | | | | **STL-10** | | | |
|---|---|---|---|---|---|---|---|---|
| | Inception Score ↑ | FID ↓ | FID Gap | Model size | Inception Score ↑ | FID ↓ | FID Gap | Model size |
| Real Dataset | 11.24±0.19 | 5.18 | | | 24.45±0.41 | 5.34 | | |
| **Standard CNN** | | | | | | | | |
| WGAN-GP | 6.68±0.06 | 39.66 | | | 8.11±0.09 | 55.64 | | |
| JS-GAN | 6.27±0.10 | 49.13 | 15.34 | 100% | 8.01±0.07 | 50.38 | 2.16 | 100% |
| RS-GAN | 7.02±0.07 | 33.79 | | | 7.62±0.08 | 52.54 | | |
| JS-GAN+ SN | 7.42±0.08 | 28.07 | 0.91 | 100% | 8.32±0.10 | 44.06 | 0.18 | 100% |
| RS-GAN+ SN | 7.32±0.08 | 27.16 | | | 8.29±0.13 | 43.88 | | |
| JS-GAN+SN; GD channel/2 | 6.85±0.08 | 33.90 | 1.16 | 29.0% | 7.69±0.05 | 57.16 | 4.69 | 32.9% |
| RS-GAN+SN; GD channel/2 | 6.74±0.04 | 32.74 | | | 7.95±0.10 | 52.47 | | |
| JS-GAN + SN; GD channel/4 | 5.83±0.07 | 52.63 | 7.26 | 9.2% | 6.90±0.06 | 72.96 | 9.35 | 11.9% |
| RS-GAN + SN; GD channel/4 | 5.94±0.09 | 45.37 | | | 7.27±0.11 | 63.61 | | |
| **ResNet** | | | | | | | | |
| JS-GAN+ SN | 8.12±0.14 | 20.13 | 0.82 | 100% | 8.87±0.07 | 36.33 | 1.56 | 100% |
| RS-GAN + SN | 7.92±0.13 | 19.31 | | | 8.96±0.10 | 34.77 | | |
| JS-GAN + SN; GD channel/2 | 7.67±0.04 | 23.29 | 1.51 | 27.5% | 8.45±0.05 | 44.39 | 2.21 | 29.0% |
| RS-GAN + SN; GD channel/2 | 7.63±0.07 | 21.78 | | | 8.47±0.09 | 42.18 | | |
| JS-GAN + SN; GD channel/4 | 6.65±0.06 | 45.20 | 13.94 | 10.4% | 8.21±0.12 | 53.57 | 1.48 | 9.2% |
| RS-GAN+ SN; GD channel/4 | 7.08±0.05 | 31.26 | | | 8.46±0.11 | 52.09 | | |
| JS-GAN + SN; BottleNeck | 7.60±0.07 | 26.98 | 1.54 | 16.8% | 8.29±0.05 | 50.38 | 3.80 | 19.2% |
| RS-GAN+ SN; BottleNeck | 7.57±0.09 | 25.44 | | | 8.52±0.11 | 46.58 | | |

Table 2: Inception score (IS) (higher is better) and Frechét Inception distance (FID) (lower is better) for JS-GAN, WGAN-GP and RS-GAN on CIFAR-10 and STL-10. We also show FID gap between JS-GAN and RS-GAN, and show the relative model size of narrow nets vs. regular nets ("regular": CNN and ResNet of [67]).

## 6   Real Data Experiments

RpGANs have been tested by Jolicoeur-Martineau [41], and are shown to be better than their SepGAN counterparts in a variety of settings[5]. In addition, RpGAN and its variants have been used in super-resolution (ESRGAN) [85] and a few recent GANs [87, 13]. Therefore, the effectiveness of RpGANs has been justified to some extent. We do not attempt to re-run the experiments merely for the purpose of justification. Instead, our goal is to use experiments to support our landscape theory.

Based on the discussions in Sec. 2, Sec. 4 and Sec. 5, we conjecture that RpGANs have a bigger advantage over SepGAN (A) with narrow deep nets, (B) in high resolution image generation, (C) with imbalanced data. Finally, (D) there exists some bad initial $D$ that makes SepGANs much worse than RpGANs. In the main text, we present results on the logistic loss (i.e., JS-GAN and RS-GAN). Results on other losses are given in the appendix.

**Experimental setting.** For setting (A), we test on CIFAR-10 and STL-10 data. For the optimizer, we use Adam with the discriminator's learning rate 0.0002. For CIFAR-10 on ResNet, we set $\beta_1 = 0$ and $\beta_2 = 0.9$ in Adam; for others, $\beta_1 = 0.5$ and $\beta_2 = 0.999$. We tune the generator's learning rate and run $100k$ iterations in total. We report the Inception score (IS) and Frechét Inception distance (FID). IS and FID are evaluated on $50k$ and $10k$ samples respectively. More details of the setting are shown in Appendix E.1, and the experimental settings for other cases besides (A) are shown in the corresponding parts in the appendix. Generated images are shown in Appendix F.

**Regular architecture and effect of spectral norm (SN).** We use the two neural architectures in [67]: standard CNN and ResNet, and report results in Table 2. First, without spectral normalization (SN),

RS-GAN achieves much higher accuracy than JS-GAN and WGAN-GP on CIFAR-10. Second, with SN, RS-GAN achieves 1-2 points lower FID score than JS-GAN, i.e., it's slightly better. We suspect that SN smoothens the landscape, thus greatly reducing the gap between JS-GAN and RS-GAN. Note that the scores of JS-GAN and WGAN-GP (both without and with SN) are comparable to or better than the scores in Table 2 of Miyato et al. [67].

**Narrow nets.** For both CNN and ResNet, we reduce the number of channels for all convolutional layers in the generator and discriminator to (1) half, (2) quarter and (3) bottleneck (for ResNet structure). The experimental results are provided in Table 2. We consider the gap between RS-GAN and JS-GAN for regular width as a baseline. For narrow nets, the gap between RS-GAN and JS-GAN is similar or larger in most cases, and can be much larger (e.g. $> 13$ FID) in some cases. The fluctuations in the gaps are consistent with landscape theory: if JS-GAN training gets stuck at a bad basin then the performance is bad; if it converges to a good basin, then the performance is reasonably good. In CIFAR-10, compared to SN-GAN with the conventional ResNet (FID=20.13), we can achieve a relatively close result by using RS-GAN with 28% parameters (half channel, FID=21.78).

**High resolution data experiments.** Sec. 2 discusses that the non-convexity of JS-GAN will become a more severe issue when the number of samples is limited compared to the data space (e.g., high resolution space or limited data points). We conduct experiments with LSUN Church and Tower images of size $256 \times 256$. RS-GAN can generate higher visual quality images than JS-GAN (Appendix F). Similarly, using another model architecture, [41] achieves a better FID score with RSGAN on the CAT dataset, which contains a small number of images (e.g., 2k $256 \times 256$ images).

**Imbalanced data experiments.** For imbalanced data, we find more evidence for the existence of JS-GAN's bad basins.The reason: JS-GAN would have a deeper bad basin, and hence a higher chance to get stuck. We conduct ablation experiments on 2-cluster data and MNIST. Both cases show that JS-GAN ends up with mode collapse while RS-GAN can generate data with proportions similar to the imbalanced true data. Check Appendix C for more.

**Bad initial point experiments.** A better landscape is more robust to initialization. On MNIST data, we find a discriminator (not random) which permits RS-GAN to converge to a much better solution than JS-GAN when used as the starting point. The FID scores are reported in the table to the right. The gap is at least 30 FID scores (a much higher gap than the gap for a random initialization). Check Appendix D for more.

| | 5e-07 | 1e-06 | 5e-06 | 1e-05 | 5e-05 | 1e-04 |
|---|---|---|---|---|---|---|
| JS-GAN | 65 | 78 | 60 | 93 | 139 | 137 |
| RSGAN | 29 | 30 | 30 | 26 | 32 | 56 |

generator lr = discriminator lr

**Combining with EMA.** It is known that non-convergence can be alleviated via EMA [88], and our theory predicts that the global landscape issue can be alleviated by RpGAN. Non-convergence and global landscape are orthogonal: no matter whether iterates are near a sub-optimal local basin or a globally-optimal basin, the algorithm may cycle. Therefore, we conjecture that the effect of EMA and the effect of RS-GAN are "additive". Our simulations show that EMA can improve both JS-GAN and RS-GAN, and the gap is approximately preserved after adding EMA. Combining EMA and RS-GAN, we achieve a similar result to the baseline (JS-GAN + SN, no EMA, FID = 20.13) using 16.8% parameters (Resnet with bottleneck plus EMA, FID=21.38). See Appendix E.1 for more.

**General RpGAN:** We conduct additional experiments on other losses, including hinge loss and least squares loss. See Appendix E.2 and E.3 for more.

## 7   Conclusion

Global optimization landscape, together with statistical analysis and convergence analysis, are important theoretical angles. In this work, we study the global landscape of GANs. Our major questions are: (1) Does the original JS-GAN formulation have a good landscape? (2) If not, is there a simple way to improve the landscape in theory? (3) Does the improved landscape lead to better performance? First, studying the empirical versions of SepGAN (extension of JS-GAN) we prove that it has exponentially many bad basins, which are mode-collapse patterns. Second, we prove that a simple coupling idea (resulting in RpGAN) can remove bad basins in theory. Finally, we verify a few predictions based on the landscape theory, e.g., RpGAN has a bigger advantage over SepGAN for narrow nets. We hope the study of the loss landscape of GANs can facilitate future optimization analysis of GANs, and help demystify the training process of GANs.

## Acknowledgements

This work is supported in part by NSF under Grant # 1718221, 2008387, 1755847 and MRI #1725729, and NIFA award 2020-67021-32799. We thank Sewoong Oh for pointing out the connection of the earlier version of our work to [41].

## Broader Impact

Generative adversarial nets (GANs) are an important tool for modeling of high-dimensional distributions. However, the theoretical understanding of GANs is still limited. This paper is a first step to add theory about the global landscape of GANs. We think this research will have a societal impact as it enables practitioners to make a more informed decision about the type of loss function that should be optimized. For example, we show that RS-GAN has benefits: (1) fewer bad basins, permitting a more stable optimization; (2) better results for narrow deep net generators, permitting its use on smaller devices, and promoting the development of smart devices and smart home services; (3) better performance on high-resolution image generation, which can be helpful in the fashion, animation, film and television industries.

Since we focus on the optimization of GANs, we do not think this research has any ethical disadvantages beyond those of GANs. Illegal fake images or videos may be the main concern related to GANs.

## Footnotes

[2] In fact, we proposed this loss in a first version of this paper, but later found that [41, 42] considered the same loss. We adopt their name RpGAN from [42].

[3]Our motivation of considering RpGAN because it breaks locality, thus possibly admitting a better landscape. This motivation is somewhat different from Jolicoeur-Martineau [41, 42].

[4]Both may be called mode collapse. Here we differentiate "mode collapse" and "mode dropping".

[5]That paper tested a number of variants, and some of them are not directly covered by our results.

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
