[Supplementary Material]

# Appendix: Towards a Better Global Loss Landscape of GANs

The code is available at https://github.com/AilsaF/RS-GAN. This appendix consists of additional experiments, related work, proofs, other results and various discussions.

# Contents

# A   Related Work

We provide a more detailed overview of related work in this section.

**Global analysis in supervised learning.** Recently, global landscape analysis has attracted much attention. See Sun [81], Sun et al. [80], Bianchini and Gori [15] for surveys and [55, 57, 26, 56, 38, 2, 92, 27] for some recent works. It is widely believed that wide networks have a nice loss landscape and thus local minima are less of a concern (e.g., [60, 32, 50]). However, this claim only holds for supervised learning, and it is not clear whether local minima cause training difficulties for GANs.

**Single-mode analysis.** For single-mode data, Feizi et al. [30] and Mescheder et al. [65] provide a global analysis of GANs. They consider a single point 0 and a single Gaussian respectively. Feizi et al. [30] differs from ours in a few aspects. First, they consider the single-mode setting which does not have an issue of mode collapse. Second, they assume $p_{\text{data}}$ is a Gaussian distribution, while we consider an arbitrary empirical distribution. Third, they analyze "quadratic-GAN," which is not common in practice, while we analyze commonly used GAN formulations (including JS-GAN).

**Mode collapse.** Mode collapse is one of the major challenges for GANs which received a lot of attention. There are a few high-level hypotheses, such as improper loss functions [3, 5] and weak discriminators [66, 78, 5, 52]. Interestingly, RpGAN both changes the loss function and improves the discriminator. The theoretical analysis of mode collapse is relatively scarce. Lin et al. [58] makes a key observation that two distributions with the same total variation (TV) distance to true distribution do not exhibit the same degree of mode collapse. They proposed to pack the samples (PacGAN) to alleviate mode collapse. This work is rather different from ours. First, they analyze the TV distance, while we analyzed SepGANs and RpGANs. Second, their analysis is statistical, while our analysis is about optimization. As for the empirical guidance, RpGAN and PacGAN are complimentary and can be used together (suggested by the author of [41]). There are a few more works that discuss mode collapse and/or local minima; we defer the discussion to Appendix A.1.

**Theoretical studies of loss functions.**   The early work on GANs [35] built a link between the min-max formulation and the J-S distance to justify the formulation. Arjovsky and Bottou [3] pointed out some possible drawbacks of J-S distance, and proposed a new loss based on Wasserstein distance, referred to as WGAN. Later, Arora et al. [5] point out that both Wasserstein distance and J-S distance are not generalizable, but they also argued that this is not too scary since people are not directly minimizing these two distances but a class of metrics referred to as "neural-network distance."

**Convergence analysis.** Many recent works analyze convergence of GANs and/or min-max optimization, e.g., [23, 22, 6, 34, 64, 88, 39, 79, 90]. These works often only analyze local stability or convergence to local minima (or stationary points), making it different from our work. Lei et al. [48] studied the convergence of WGAN, but restricted to 1-layer neural nets.

**Other theoretical analysis.** There are a few other theoretical analysis of GANs, e.g., [68, 59, 29, 16, 8, 51, 61, 48]. Most of these works are not directly related to our work.

**Other GAN Variants.** There are many GAN variants, e.g., WGAN [4, 3, 36] and variants [86, 46, 1, 24, 25], $f$-GAN [74], SN-GAN [67], self-attention GAN [89], StyleGAN [43, 44] and many more [63, 69, 12, 70, 21, 54, 49, 78, 74, 75, 66, 37, 76, 10, 49]. Our analysis framework (analyzing global landscape of empirical loss) can potentially be applied to more variants mentioned above.

## A.1   Related Works on Local Minima and Mode Collapse

We discuss a few related works on local minima and mode collapse, including Kodali et al. [45], Li and Malik [53] and Unterthiner et al. [83] that are mentioned in the main text.

**DRAGAN.** Kodali et al. [45] suggested the connection between mode collapse and a bad equilibrium based on the following empirical observation: a sudden increase of the gradient norm of the discriminator during training is associated with a sudden drop of the IS score. However, Kodali et al. [45] don't present formal theoretical results on the relation between mode collapse and a bad equilibrium.

**IMLE.** Li and Malik [53] proposed implicit maximum likelihood estimation (IMLE). The empirical version of IMLE in the parameter space is the following:

$$\min_{w} \sum_{j=1}^{n} \min_{i \in \{1,\dots,m\}} \|x_i - G_w(z_j)\|^2. \tag{9}$$

In other words, for each generated sample $y_j = G_w(z_j)$, the loss is the distance from $y_j$ to the closest true sample $x_i$. Interestingly, IMLE and RpGAN both couple the true data and the fake data in the loss. The differences are two fold: first, IMLE does not have an extra discriminator $f_\theta$, while RpGAN has; second, IMLE compares $y_j$ with all $x_i$ (so as to find the nearest neighbor) while RpGAN compares $y_j$ with an arbitrary $x_j$. See Table 3 for a comparison. Note that Li and Malik [53] don't present formal theoretical results on the landscape.

Table 3: Models that couple true data and fake data in the loss

| Model name | Empirical form of loss [i] | Form of coupling | Optimization |
|---|---|---|---|
| RpGAN [41] | $\max_f \sum_j h(f(x_j) - f(y_j))$ | pairing | min-max [ii] |
| RaGAN [iii] [41] | $\max_f \sum_j h(\frac{1}{n} \sum_{i=1}^{n} f(x_i) - f(y_j))$ | comparing with average | min-max |
| (max-)sliced-WGAN [24, 25] | $\max_{\|f\|_L \le 1} \sum_{i=1}^{n} [f(X)_{(i)} - f(Y)_{(i)}]^2$ [iv] | pairing sorted output | min-max |
| IMLE [53] | $\sum_j \min_{i \in [n]} \|y_j - x_i\|^2$ | comparing with closest | min |
| Coulomb-GAN [83] | $\sum_{i,j} k(x_i, x_j) + \sum_{i,j} k(y_i, y_j) - 2 \sum_{i,j} k(x_i, y_j)$ [v] | all-pairs | non-zero-sum game [vi] |

[i] We show the empirical form of the loss in the function space. Rigorously speaking, the provided form is the the loss for one mini-batch; in practice, in different iterations of SGD we will use different samples of $x_i, y_j$. For the empirical loss in the parameter space, we shall replace $f$ by $f_\theta$ and $y_j$ by $G_w(z_j)$.    [ii] Besides the zero-sum game form (min-max form), RpGAN can be easily modified to a non-zero-sum game form ("non-saturating version" proposed in [35]).    [iii] The precise expression of RaGAN (relativistic averaging GAN) shall be $\sum_j h_1(\frac{1}{n} \sum_{i=1}^{n} f_\theta(x_i) - f_\theta(y_j)) + \sum_i h_2(\frac{1}{n} \sum_{j=1}^{n} f_\theta(y_j) - f_\theta(x_i))$, but for simplicity we only present one term in the table. [iv] Here $f(X)_{(1)} \le \cdots \le f(X)_{(n)}$ and $f(Y)_{(1)} \le \cdots \le f(Y)_{(n)}$ are the sorted versions of $f(x_i)$'s and $f(y_i)$'s respectively. [v] Here $k$ is the Coulomb kernel, defined as $k(u,v) = \frac{1}{(\sqrt{\|u-v\|^2 + \epsilon^2})^\alpha}$ where $u, v \in \mathbb{R}^d$, $\alpha \le d - 2$ and $\epsilon > 0$. The original form of Coulomb-GAN is a non-zero-sum game, but it is straightforward to transfer the formulation to a pure minimization form since the discriminator-minimization problem has a closed form solution (used in the proof of [83, Theorem 2]). We presented the transformed minimization problem here.    [vi] Coulomb-GAN is presented as a non-zero-sum game, but as mentioned earlier it can be transformed to a minimization problem. The original Coulomb-GAN uses a smoothing operator in the generator loss; in this empirical form, we omit the smoothing operator for easier comparison (thus it is not the same as Coulomb-GAN). In the table, we show the resulting loss in the pure minimization form. Unlike SepGAN and RpGAN that can be written as either min-max form or non-zero-sum game form, we point out that there is no min-max form for Coulomb-GAN, since the design principle of Coulomb-GAN is very different from typical GANs.

**Coulomb-GAN.** Unterthiner et al. [83] argued that mode collapse can be a local Nash equilibrium in an example of two clusters (see [83, Appendix A.1]). They further proposed ColumbGAN and claimed that every local Nash equilibrium is a global Nash equilibrium (see [83, Theorem 2]). Their study is different from ours in a few aspects. **First**, they still consider the pdf $p_g$, though restrict the possible movement of $p_g$ (according to a continuity equation). In contrast, we consider the empirical loss in particle space. **Second**, the bad landscape of JS-GAN is discussed in words for the 2-cluster case [83, Appendix A.1], but not formally proved. In contrast, we prove rigorous result for the general case. **Third**, they do not study parameter space (though with informal discussion). **Fourth**, they do not present landscape-related experiments, such as the narrow-net experiments we have done.

**Common idea: Coupling true data and fake data.** Interestingly, similar to IMLE and RpGAN, ColumbGAN also coupled the true data and fake data in the loss functions. RpGAN, RaGAN (a variant of RpGAN considered in [41]), IMLE and ColumbGAN differ in two aspects: the specific form of coupling (pairing, comparing with average, comparing with the closest, all possible pairs), and the specific form of optimization (pure minimization, min-max, non-zero-sum game). See the comparison in Table 3. It is interesting that all three lines of work choose to couple true data and fake data to resolve the issue of mode collapse. We suspect it is hard to prove similar results on the landscape of empirical loss for IMLE and Coulomb-GAN.

**Relation to (max)-sliced Wasserstein GAN.** We point out that the sliced Wasserstein GAN (sliced-WGAN) [24] and the max-sliced Wasserstein GAN (max-sliced-WGAN) [25] also couple the true data and fake data. For any function $f$, denote $f(X) = (f(x_1), \ldots, f(x_n))$ and $f(Y) = (f(y_1), \ldots, f(y_n))$. The empirical version of the max-sliced Wasserstein GAN can be written as

$$\min_Y \max_{\|f\|_L \le 1} W_2(f(X), f(Y))^2. \tag{10}$$

Here $f$ is a neural net with codomain $\mathbb{R}$, and $W_2$ is the Wasserstein-2-distance. Denote $f(X)_{(1)} \le \cdots \le f(X)_{(n)}$ and $f(Y)_{(1)} \le \cdots \le f(Y)_{(n)}$ as the sorted versions of $f(x_i)$'s and $f(y_i)$'s respec-

(a) JS-GAN 1st run     (b) JS-GAN 2nd run     (c) RS-GAN 1st run     (d) RS-GAN 2nd run

Figure 7: Comparison of JS-GAN and RS-GAN for two different runs. First row: D loss; second row: fake data movement during training.

tively. Then Eq. (10) is equivalent to

$$(\text{max-})\text{sliced-WGAN}^6 : \min_Y \max_{|f|_L \le 1} \sum_{i=1}^n [f(X)_{(i)} - f(Y)_{(i)}]^2. \tag{11}$$

This form is quite close to RpGAN (when $h(t) = t^2$): the only differences are the sorting of $f(X), f(Y)$ and the extra constraint $|f|_L \le 1$. The extra constraint $|f|_L \le 1$ is due to unbounded $h$, and can be removed if we use an upper bounded $h$ (which leads to a sorting version of RpGAN). See the comparison of max-sliced-WGAN with RpGAN and other models in Table 3.

**Nash equilibria for Gaussian data.** A very recent work Farnia and Ozdaglar [28] shows that for a non-realizable case (with a linear generator) Nash equilibria may not exist for learning a Gaussian distribution. This setting is quite different from ours.

## B  2-Cluster Experiments: Details and More Discussions

In this part, we present details of the experiments in Section 5 and other complementary experiments.

**Experimental Setting.** The code is provided in "GAN_2Cluster.py". We sample 100 points from two clusters of data near 0 and 4 (roughly 50 in each cluster). We use GD with momentum parameter 0.9 for both $D$ and $G$. The default learning rate is (Dlr, Glr) = $(10^{-2}, 10^{-2})$. The default inner-iteration-number for the discriminator and the generator are (DIter, GIter) = $(10, 10)$. The discriminator and generator net are a 4-layer network (with 2 hidden layers) with sigmoid activation and tanh activation respectively. The default neural network width (Dwidth, Gwidth) = $(10, 5)$. We will also discuss the results of other hyperparameters. The default number of training iterations is MaxIter = 5000. We use the non-saturating versions for both JS-GAN and RS-GAN.

**Understanding the effect of mode collapse, by checking D loss evolution and data movement.** In the main text, we discussed that mode collapse can slow down training of JS-GAN. For easier understanding of the training process, we add the visualization of the data movement (which is possible since we are dealing with 1-dimensional data) in Figure 7. We use the y-axis to denote the data position, and x-axis to denote the iteration. The blue curves represent the movement of all fake data during training, and the red straight lines represent the position of true data (two clusters). The training time may vary across different runs, but overall the time for JS-GAN is about 2-4 times longer than that for RS-GAN.

**Effect of width.** The default width is (Dwidth, Gwidth) = $(10, 5)$. We tested two other settings: $(20, 10)$ and $(5, 3)$. For the wide-network setting, the convergence of both JS-GAN and RS-GAN are much faster, but RS-GAN is still faster than JS-GAN in most cases; see Fig. 8. For the narrow-network setting, RS-GAN can recover two modes in all five runs, while JS-GAN fails in two of the five runs (within 5k iterations). See Fig. 9 for one success case of JS-GAN and one failure case of JS-GAN. In the failure case, JS-GAN completely gets stuck at mode collapse, and the $D$ loss is stuck at around 0.48, consistent with our theory.

(a) JS-GAN 1st run     (b) JS-GAN 2nd run     (c) RS-GAN 1st run     (d) RS-GAN 2nd run

Figure 8: Wide network (Dwidth, Gwidth) $= (20, 10)$: JS-GAN and RS-GAN in two different runs. Compare to regular widths (Dwidth, Gwidth) $= (10, 5)$, both GANs converge faster. Anyhow, RS-GAN is still 2-3 times faster than JS-GAN.

(a) JS-GAN 1st run     (b) JS-GAN 2nd run     (c) RS-GAN 1st run     (d) RS-GAN 2nd run

Figure 9: Narrow network setting: Comparison of JS-GAN and RS-GAN in two runs. RS-GAN is a few times faster than JS-GAN in general. Compare to default widths (D width 10, G width 5), both GANs converge slower. In one case (b), JS-GAN gets stuck at mode collapse.

**Other hyperparameters.** Besides the width, the learning rates and (DIter, GIter) will also affect the training process. As for (DIter, GIter), we use $(10, 10)$ as default, but other choices such as $(5, 2)$ and $(1, 1)$ also work. As for learning rates, we use $(0.01, 0.01)$ as default, but smaller learning rates such as $(0.001, 0.001)$ also work. Different from the default hyper-parameters, for some hyper-parameters, the D loss of JS-GAN does not reach $0.48$, indicating that the basin only attracts the iterates half-way. Nevertheless, in most settings RS-GAN is still faster than JS-GAN.

## C    Result and Experiments for Imbalanced Data Distribution

In the main results, we assume $x_i$'s are distinct. In this section, we allow $x_i$'s to be in general positions, i.e., they can overlap. The 2-point model can only approximate two balanced clusters. Allowing $x_i$'s to overlap, we are able to analyze imbalanced two clusters. We will show: (i) a theoretical result for 2-cluster data; (ii) experiments on imbalanced 2-cluster data and MNIST.

### C.1    Imbalanced Data: Math Results for Two-Clusters

Assume there are $n$ true data points $X = (x_1, \ldots, x_n)$ in two modes with proportion $\alpha$ and $1 - \alpha$ respectively, where $\alpha > 0.5$. More precisely, assume $x_1 = x_2 = \cdots = x_{n\alpha}$ and $x_{n\alpha+1} = \cdots = x_n$, and denote two multi-sets $\mathcal{X}_1 = \{x_1, x_2, \ldots, x_{n\alpha}\}$ and $\mathcal{X}_2 = \{x_{n\alpha+1}, x_2, \ldots, x_n\}$. Denote $Y = (y_1, \ldots, y_n)$ as the tuple of all generated points, and let $\mathcal{Y}$ be the multiset $\{y_1, \ldots, y_n\}$.

**Claim C.1.** *Consider the JS-GAN loss defined in Eq. (1), where $X$ is defined above. We have*

$$\phi_{\mathrm{JS}}(Y, X) = q_\alpha(m_1) + q_{1-\alpha}(m_2), if \, |\mathcal{X}_1 \cap \mathcal{Y}|=m_1, |\mathcal{X}_2 \cap \mathcal{Y}|=m_2,$$

$$where \; q_\alpha(m) \triangleq \frac{\alpha}{2} \log(\alpha n) + \frac{m}{2n} \log m - \frac{\alpha n + m}{2n} \log(\alpha n + m). \tag{12}$$

*As a result, the global minimal loss is $-\log 2$, which is achieved iff $\mathcal{Y} = \mathcal{X}_1 \cup \mathcal{X}_2$.*

**Corollary C.1.** *Suppose $\hat{Y} = (\hat{y}_1, \ldots, \hat{y}_n)$ satisfies $|\mathcal{X}_1 \cap \hat{\mathcal{Y}}| = n_1, |\mathcal{X}_2 \cap \hat{\mathcal{Y}}| = n - n_1$, where $\hat{\mathcal{Y}} = \{\hat{y}_1, \ldots, \hat{y}_n\}$ is the multiset of all $\hat{y}_j$'s, then $\hat{Y}$ is a strict local minimum. Moreover, if $n_1 \neq n\alpha$, then $\hat{Y}$ is a sub-optimal strict local minimum.*

The proofs of Claim C.1 and Corollary C.1 are given in Appendix H.3.

Denote $m_1 \triangleq |\mathcal{X}_2 \cap \mathcal{Y}|, m_2 \triangleq |\mathcal{X}_1 \cap \mathcal{Y}|$. The value $q_\alpha(n)$ indicates the value of $\phi(Y, X)$ at the mode collapsed pattern (state 1a) where $m_1 = n, m_2 = 0$. Note that $q_\alpha(n) = \frac{\alpha}{2} \log \frac{\alpha}{\alpha+1} + \frac{1}{2} \log \frac{1}{\alpha+1}$ is a strictly decreasing function of $\alpha$. When $\alpha = 1/2$, $q_\alpha(n) = \frac{1}{4} \log \frac{1}{3} + \frac{1}{2} \log \frac{2}{3} \approx -0.4774$; when $\alpha = 2/3$, $q_\alpha(n) \approx -0.5608$. The more imbalanced the data are (larger $\alpha$), the smaller $q_\alpha(n)$, and

Figure 10: Illustration of the landscape of JS-GAN for balanced two clusters with $\alpha = 0.5$ (left) and imbalanced two clusters with $\alpha = 2/3$ (right). Denote $m_i \triangleq |\mathcal{X}_i \cap \mathcal{Y}|, i = 1, 2$. Here state 0, state 1a, state 1b, state 2 represent $(m_1, m_2) = (0, 0), (n\alpha, 0), (n\alpha, 0), (n\alpha, n(1 - \alpha))$ respectively. By Claim C.1, for $\alpha = 1/2$, $q_\alpha(n) \approx -0.48$ and $q_\alpha(\alpha n) \approx -0.35$; for $\alpha = 2/3$, $q_\alpha(n) \approx -0.56$ and $q_\alpha(\alpha n) \approx -0.46$. Different from the 2-point-case landscape in Fig 5, there should be some intermediate patterns (satisfying $m_1 \leq n, m_2 = 0$), but for simplicity we do not show them. From state 1a to state 2, $Y$ can go through state 1b or go through state 0, but we only show the path through state 1b. We view the gap between state 0 and state 1a as an approximation of the "depth" of the basin.

| (a) JS-GAN D loss | (b) JS-GAN Data Evolution | (c) RS-GAN D loss | (d) RS-GAN Data Evolution |

Figure 11: Imbalanced 2-cluster result: comparison of JS-GAN in (a) and (b), and RS-GAN in (c) and (d). (a) and (c): evolution of D loss; (b) and (d): data position movement during training.

further the deeper the basin. In Figure 10, we compare the loss landscape of the balanced case $\alpha = 1/2$ and the imbalanced case $\alpha = 2/3$.

We suspect that the deeper basin in the imbalanced case will make it harder to escape mode collapse for JS-GAN. We then make the following prediction: for JS-GAN, mode collapse is a more severe issue for imbalanced data than it is for balanced data. For RS-GAN, the performance does not change much as data becomes more imbalanced. We will verify this prediction in the next subsections.

## C.2 Experiments

**2-Cluster Experiments.** For the balanced case, the experiment is described in Appendix B. Both JS-GAN and RS-GAN can converge to the two-mode-distribution. For the imbalanced case where $\alpha = \frac{2}{3}$, with other hyper-parameters unchanged, JS-GAN falls into mode collapse while RS-GAN generates the true distribution (2/3 in mode 1 and 1/3 in mode 2) (see Fig. 11). The loss $\phi_{\text{JS}}(Y, X)$ ends up at approximately -0.56, which matches Claim C.1.

**MNIST experiments.** To ease visualization, we create an MNIST sub-dataset only containing 5's and 7's. We use the CNN structure of Tab. 7 and train for $30k$ iterations. For the balanced case, the number of 5's and 7's are identical (i.e., ratio 1:1). Both JS-GAN and RS-GAN generate a roughly equal number of 5's and 7's, as shown in Fig. 12(a,b). For the imbalanced case with 4 times more 7's than 5's (ratio 1:5), JS-GAN only generates 7's, while RS-GAN generates 13 5's among 64 generated samples, aligning with the true data distribution (see Fig. 12(c,d)).

The above two experiments verify our earlier prediction that RS-GAN is robust to imbalanced data while JS-GAN easily gets stuck at mode collapse for imbalanced data.

## D Experiments of Bad Initialization

A bad optimization landscape does not mean the algorithm always converges to bad local minima[7]. A 'bad' landscape means is that there exists a "bad" initial point (the blue point in Fig. 13(a)) that it will lead to a 'bad' final solution upon training. In contrast, a good landscape is more robust to the initial point: starting from any initial point (e.g., two points shown in Fig. 13(b)), the algorithm can still find a good solution. Therefore, bad optimization landscape of JS-GAN does not mean the performance of JS-GAN is bad for *any* initial point, but it should imply that JS-GAN is bad for *certain* initial points.

Next, we will show experiments that support this prediction.

| (a) balanced MNIST: JS-GAN | (b) balanced MNIST: RS-GAN | (c) imbalanced MNIST: JS-GAN | (d) imbalanced MNIST: RS-GAN |

Figure 12: Balanced and Imbalanced MNIST setting: Comparison of JS-GAN and RS-GAN.

**5-Gaussian Experiments**. We consider a 2-dimensional 5-Gaussian distribution as illustrated in Fig. 14(a). We design a procedure to find an initial discriminator and generator. For JS-GAN or RS-GAN, in some runs we obtain mode collapse and in some runs we obtain perfect recovery. Firstly, for the runs achieving perfect recovery (Fig. 14(b)) in JS-GAN and RS-GAN respectively, we pick the generators at the converged solution, which we denote as $G_{JS0}$ and $G_{RS0}$ respectively. Secondly, for the runs attaining mode collapse (Fig. 14(c)) in JS-GAN and RS-GAN respectively, we pick the discriminators at the converged solution, referred to as $D_{JS0}$ and $D_{RS0}$, Then we re-train both JS-GAN and RS-GAN from $(D_{JS0}, G_{JS0})$ and $(D_{RS0}, G_{RS0})$ respectively.

We define an evaluation metric $\Psi = \sum_{k=1}^{K} \min_{1 \leq i \leq 10^4} (\alpha \|x_i - C_k\|)$, where $C_k$'s are the cluster centers, $\alpha$ is a scalar and $x_i$'s are $10^4$ true data samples. We repeat the experiment $S = 50$ times and compute the average $\Psi$. The larger the metric, the worse the generated points. As shown in Fig. 14(a), the metric $\Phi$ is much higher for JS-GAN than for RS-GAN, for various learning rates lr.

Figure 15: MNIST experiment

**MNIST Experiments**. We use a similar strategy to find initial parameters for MNIST data. Fig. 15 (also in Sec. 6) shows that RS-GAN generates much lower FID scores (30+ gap) than JS-GAN.

The two experiments verify our prediction that RS-GAN is more robust to initialization, which supports our theory that RS-GAN enjoys a better landscape than JS-GAN.

# E  Experiments of Regular Training: More Details and More Results

In this section, we present details of the regular experiments in Sec. 6 and a few more experiments.

## E.1  Experiment Details and More Experiments with Logistic Loss

**Non-saturating version.** Following the standard practice [35], if $\lim_{t\to\infty} h(t) = 0$, we use the non-saturating version of RpGAN in practical training:

$$\min_{\theta} L_D(\theta; w) \triangleq \frac{1}{n} \sum_i h(f_\theta(x_i)) - f_\theta(G_w(z_i))),$$

$$\min_{w} L_G(w; \theta) \triangleq \frac{1}{n} \sum_i h(f_\theta(G_w(z_i)) - f_\theta(x_i))). \tag{13}$$

For logistic and hinge loss, we use Eq. (13). For least-square loss, we use the original min-max version (check Appendix E.3 for more). We use alternating stochastic GDA to solve this problem.

**Neural-net structures:** We conduct experiments on two datasets: CIFAR-10 ($32 \times 32$ size) and STL-10 ($48 \times 48$ size) on both standard CNN and ResNet. As mentioned in Sec. 6, we also conduct experiments on the narrower nets: we reduce the number of channels for all convolutional layers in the generator and discriminator to (1) half, (2) quarter and (3) bottleneck (for ResNet structure), The

| (a) bad landscape with bad local minima | (b) good landscape with multiple global minima |

Figure 13: Left: for a bad landscape, a good initial point (red) leads to convergence to a global optima while a bad one (blue) does not. Right: for a good landscape, two initial points both converge to global minima.

Figure 14: Five Gaussian experiment. (a): ground truth. (b): generated data covers all five clusters. (c): mode collapse happens and only two clusters get covered. (d) JS-GAN and RSGAN's loss $\Psi$ under different lr (generator lr = discriminator lr).

| | CIFAR-10 | | CIFAR-10+EMA | | STL-10+EMA | |
|---|---|---|---|---|---|---|
| | IS ↑ | FID ↓ | IS ↑ | FID ↓ | IS ↑ | FID ↓ |
| **ResNet** | | | | | | |
| JS-GAN+SN | 8.03±0.10 | 20.06±0.18 | 8.41±0.09 | 17.79±0.43 | 9.14±0.12 | 33.06 |
| RS-GAN+SN | 7.94±0.09 | 19.79±0.57 | 8.37±0.10 | 17.75±0.56 | 9.23±0.08 | 31.87 |
| JS-GAN+SN+GD channel/2 | 7.77±0.08 | 23.36±0.46 | 8.24±0.08 | 20.55±0.59 | 8.69±0.08 | 42.05 |
| RS-GAN+SN+GD channel/2 | 7.76±0.07 | 21.63±0.51 | 8.21±0.09 | 18.91±0.45 | 8.77±0.13 | 39.31 |
| JS-GAN+SN+GD channel/4 | 6.75±0.06 | 44.39±4.38 | 7.18±0.06 | 38.75±6.28 | 8.42±0.06 | 52.38 |
| RS-GAN+SN+GD feature/4 | 7.20±0.07 | 31.40±0.78 | 7.60±0.06 | 26.85±0.56 | 8.43±0.10 | 48.92 |
| JS-GAN+SN+BottleNeck | 7.51±0.07 | 27.33±1.05 | 7.99±0.10 | 23.71±0.86 | 8.37±0.08 | 47.97 |
| RS-GAN+SN+BottleNeck | 7.52±0.10 | 25.05±0.35 | 8.06±0.11 | 21.29±0.22 | 8.48±0.06 | 44.60 |

Table 4: Repeat the experiments (logistic loss) in Tab. 2 with at least three seeds.

architectures are shown in Tab. 7 (CNN), Tab. 9 (ResNet for CIFAR) and Tab. 10 (ResNet for STL) and Tab. 11 (Bottleneck for CIFAR) and Tab. 12 (Bottleneck for STL).

**Hyper-parameters:** We use a batchsize of 64. For CIFAR-10 on ResNet we set $\beta_1 = 0$ and $\beta_2 = 0.9$ in Adam. For others, $\beta_1 = 0.5$ and $\beta_2 = 0.999$. We use GIter = 1 for both CNN and ResNet. We also use DIter = 1 for CNN and DIter = 5 for ResNet. We fix the learning rate for the discriminator (dlr) to be 2e-4. For RpGANs, we find that the learning rate for the generator (glr) needs to be larger than dlr to keep the training balanced. Thus we tune glr using parameters in the set 2e-4, 5e-4, 1e-3, 1.5e-3. For SepGAN, we set glr = 0.0002 for SepGANs (JS-GAN,hinge-GAN) as suggested by [67, 76] [8]. See Tab. 13 for the learning rate of RS-GAN and hyper-parameters of WGAN-GP.

**More details of EMA:** In Sec. 6, we conjectured that the effect of EMA (exponential moving average) [88] and RpGAN are additive. Suppose $w^{(t)}$ is the generator parameter in $t$-th iteration of one run, the EMA generator at the $t^{\text{th}}$ iteration is computed as follows $w_{\text{EMA}}^{(t)} = \beta w_{\text{EMA}}^{(t-1)} + (1-\beta)w^{(t)}$, where $w_{\text{EMA}}^{(0)} = w^{(0)}$. Note that EMA is a post-hoc processing step, and does not affect the training process. Intuitively, the EMA generator is closer to the bottom of a basin while the real training is circling around a basin due to the minmax structure. We set $\beta = 0.9999$. As Tab. 4 shows, while EMA improves both JS-GAN and RS-GAN, RS-GAN is still better than JS-GAN.

**Results on Logistic Loss with More Seeds:** Besides the result in Tab. 2, we run at least 3 extra seeds for all experiments with ResNet structure on CIFAR-10 to show that the results are consistent across different runs. We report the results in Tab. 4, and find RS-GAN is still better than JS-GAN and the gap increases as the networks become narrower.

**Samples of image generation:** Generated samples obtained upon training on CIFAR-10 are given in Fig. 16 for CNN, Fig. 17 for ResNet. Generated samples obtained upon training on STL-10 dataset are given in Fig. 18 for CNN, Fig. 19 for ResNet. Instead of cherry-picking, all sample images are generated from random sampled Gaussian noise.

|  | **CIFAR-10** | | | **CIFAR-10 + EMA** | | |
|---|---|---|---|---|---|---|
|  | IS ↑ | FID ↓ | FID Gap | IS ↑ | FID ↓ | FID Gap |
| **ResNet + Hinge Loss** | | | | | | |
| Hinge-GAN | 7.92±0.08 | 21.30 | | 8.44±0.10 | 17.43 | |
| Hinge-GAN +GD channel/2 | 7.63±0.05 | 27.21 | | 7.90±0.08 | 24.35 | |
| Hinge-GAN +GD channel/4 | 6.79±0.09 | 37.51 | | 7.39±0.07 | 34.45 | |
| Hinge-GAN +BottleNeck | 7.16±0.10 | 33.24 | | 7.91±0.09 | 26.56 | |
| Rp-Hinge-GAN | 7.84±0.09 | 19.10 | 2.20 | 8.21±0.09 | 17.19 | 0.24 |
| Rp-Hinge-GAN +GD channel/2 | 7.77±0.08 | 21.10 | 6.11 | 8.34±0.11 | 19.19 | 5.17 |
| Rp-Hinge-GAN +GD channel/4 | 7.21±0.11 | 29.41 | 8.10 | 7.77±0.08 | 25.57 | 8.88 |
| Rp-Hinge-GAN +BottleNeck | 7.52±0.07 | 23.28 | 9.96 | 8.05±0.07 | 22.03 | 4.53 |

Table 5: Comparison of Hinge-GAN and Rp-Hinge-GAN. We also show the FID gap between Rp-Hinge-GAN with Hinge-GAN (e.g. $2.20 = 21.30 - 19.10$ and $9.96 = 33.24 - 23.28$).

## E.2 Experiments with Hinge Loss

Hinge loss has become popular in GANs [82, 67, 18]. The empirical loss of hinge-GAN is

$$\min_\theta L_D^{\text{Hinge}}(\theta; w) \triangleq \frac{1}{2n} \left[ \sum_i \max(0, 1 - D_\theta(x_i)) + \sum_i \max(0, 1 + D_\theta(G_w(z_i))) \right],$$

$$\min_w L_G^{\text{Hinge}}(w; \theta) \triangleq -\frac{1}{n} \sum_i D_\theta(G_w(z_i)).$$

Note that Hinge-GAN applies the hinge loss for the discriminator, and linear loss for the generator. This is a variant of SepGAN with $h_1(t) = h_2(t) = -\max(0, 1 - t)$.

The Rp-hinge-GAN is RpGAN given in Eq. (13) with $h(t) = -\max(0, 1 - t)$:

$$\min_\theta L_D^{\text{R-Hinge}}(\theta; w) \triangleq \frac{1}{n} \sum_i \max(0, 1 + (f_\theta(G_w(z_i)) - f_\theta(x_i))),$$

$$\min_w L_G^{\text{R-Hinge}}(w; \theta) \triangleq \frac{1}{n} \sum_i \max(0, 1 + (f_\theta(x_i) - f_\theta(G_w(z_i)))).$$

We compare them on ResNet with the hyper-parameter settings in Appendix E.1. As Tab. 5 shows, Rp-Hinge-GAN (both versions) performs better than Hinge-GAN. For narrower networks, the gap is 4 to 9 FID scores, larger than the gap for the logistic loss.

## E.3 Experiments with Least Square Loss

We consider the least square loss. The LS-GAN [62] is defined as follows:

$$\min_\theta L_D^{\text{LS}}(\theta; w) \triangleq \frac{1}{2n} \left[ \sum_i (f_\theta(x_i) - 1)^2 + \sum_i f_\theta(G_w(z_i))^2 \right],$$

$$\min_w L_G^{\text{LS}}(w; \theta) \triangleq \frac{1}{n} \sum_i (f_\theta(G_w(z_i)) - 1)^2.$$

This is a non-zero-sum variant of SepGAN with $h_1(t) = -(1 - t)^2, h_2(t) = -t^2$.

Rp-LS-GAN addresses the following objectives:

$$\min_\theta L_D^{\text{Rp-LS}}(\theta; w) \triangleq \frac{1}{n} \sum_i (f_\theta(x_i) - f_\theta(G(z_i)) - 1)^2,$$

$$\min_w L_G^{\text{Rp-LS}}(w; \theta) \triangleq -L_D^{\text{Rp-LS}}(\theta; w) = -\frac{1}{n} \sum_i (f_\theta(x_i) - f_\theta(G_w(z_i)) - 1)^2. \tag{14}$$

For least square loss $h(t) = -(t - 1)^2$, the gradient vanishing issue due to $h$ does not exist, thus we can use the min-max version given in Eq. (14) in practice. Our version of Rp-LS-GAN is actually different from the version of Rp-LS-GAN in [41] which is similar to Eq. (13) with least square $h$.

In Tab. 6 we compare LS-GAN and Rp-LS-GAN on CIFAR-10 with CNN architectures detailed in Tab. 7. As Tab. 6 shows, Rp-LS-GAN is slightly worse than LS-GAN in regular width, but is better than LS-GAN (with 5.7 FID gap) when using 1/4 width.

## F Experiments on High Resolution Data

There are two approaches to achieve a good landscape: one uses a wide enough neural net [73, 50], and the other uses a large enough number of samples (approaching convexity of pdf space). As we

| | Regular width | | | channel/2 | | | channel/4 | | |
|---|---|---|---|---|---|---|---|---|---|
| | IS | FID | FID Gap | IS | FID | FID Gap | IS | FID | FID Gap |
| LS-GAN | 6.91±0.10 | **32.93** | | 6.63±0.08 | 37.83 | | 5.69±0.10 | 48.63 | |
| Rp-LS-GAN | **7.09**±0.07 | 34.78 | -1.85 | **6.94**±0.04 | **34.34** | 3.49 | **6.22**±0.10 | **42.86** | 5.77 |

Table 6: Comparison of LS-GAN and Rp-LS-GAN on CIFAR-10 with the CNN structure.

discuss in Sec. 2 (see also Appendix G.1), when the number of samples is far from enough for filling the data space, the convexity (of pdf space) may vanish. A higher dimension of data implies a larger gap between empirical loss and population loss, thus the non-convexity issue will become more severe. Thus we conjecture that JS-GAN suffers more for higher resolution data generation.

We consider $256 \times 256$ LSUN Church and Tower datasets with CNN architecture in Tab. 8. For RS-GAN, we set glr = 1e-3 and dlr = 2e-4 We train $100,000$ iterations with batchsize $64$. The generated images are presented in Fig. 20. For both datasets, RS-GAN outperforms JS-GAN visually.

# G   Discussions on Empirical Loss and Population Loss (complements Sec. 2)

As mentioned in Sec. 2, the pdf space view (the population loss) was first used in [35], and became quite popular for GAN analysis. See, e.g., [71, 40, 20]. In this part, we provide more discussions on the relation of empirical loss and population loss in GANs.

## G.1   Particle space or probability space?

Suppose $p_z = \mathcal{N}(0, I_{d_z})$ (or other distributions) is the distribution of the latent variable $z$, and $Z = (z_1, \ldots, z_n)$ are the samples of latent variables. During training, the parameter $w$ of the generator net $G_w$ is moving, and, as a result, both the pdf $p_g = G_w(p_z)$ and the particles $y_j = G_w(z_j)$ move accordingly. Therefore, GAN training can be viewed as either probability space optimization or particle space optimization. The two views (pdf space and particle space) are illustrated in Figure 1.

In the probability space view, an implicit assumption is that the pdf $p_g$ moves *freely*; in the particle space view, we assume the particles move *freely*. Free-particle-movement implies free-pdf-movement if the particles almost occupy the whole space (a one-mode distribution), as shown in Fig. 21. However, for multi-mode distributions in high-dimensional space, the particles are sparse in the space, and free-particle-movement does NOT imply free-pdf-movement. This gap was also pointed out in [83]; here, we stress that the gap becomes larger for sparser samples (eiher due to few samples ore high dimension). This forms the foundation for experiments in App. F.

To illustrate the gap between free-pdf-movement and free-particle-movement, we use an example of learning a two-mode distribution $p_{\text{data}}$. Suppose we start from an initial two-mode distribution $p_{\text{g}}$, as shown Figure 22. To learn $p_{\text{data}}$, we need to do two things: first, move the two modes of $p_{\text{g}}$ to roughly overlap with the two modes of $p_{\text{data}}$ which we call "macro-learning"; second, adjust the distributions of each mode to match those of $p_{\text{data}}$, which we call "micro-learning." This decomposition is illustrated in Fig. 22 and 23. In micro-learning, the pdf can move freely, but in macro-learning, the whole mode has to move together and cannot move freely in the pdf space.

Figure 21: Illustration of the learning process of the single mode. The generated samples are moving, which corresponds to adjustment of the probability densities.

Figure 22: Illustration of the process of learning a multi-mode distribution. We decompose this process into two parts in the next figure.

(a) Macro-learning

(b) Micro-learning

Figure 23: Decomposing learning a multi-mode distribution into macro-learning and micro-learning. Macro-learning refers to the movement of the whole mode towards the underlying data mode. Micro-learning refers to the adjustment of the distribution within each mode. If macro-learning fails, then an entire mode is missed in the generated distributions, which corresponds to mode collapse.

## G.2 Empirical loss and population loss

The population version of RpGAN [41] is $\min_{p_{\text{data}}} \phi_{\text{R,E}}(p_{\text{g}}, p_{\text{data}})$, where

$$\phi_{\text{R,E}}(p_{\text{g}}, p_{\text{data}}) = \sup_{f \in C(\mathbb{R}^d)} \mathbb{E}_{(x,y) \sim (p_{\text{g}}, p_{\text{data}})}[h(f(x) - f(y))]. \tag{15}$$

Suppose we sample $x_1, \ldots, x_n \sim p_{\text{data}}$ and $y_1, \ldots, y_n \sim p_{\text{g}}$, then $\frac{1}{n} \sum_{i=1}^n [h(f(x_i) - f(y_i))]$ is an approximation of $\mathbb{E}_{(x,y) \sim (p_{\text{g}}, p_{\text{data}})}[h(f(x) - f(y))]$. The empirical version of RpGAN addresses $\min_{Y \in \mathbb{R}^{d \times n}} \phi_{\text{R}}(Y, X)$, where

$$\phi_{\text{R}}(Y, X) = \sup_{f \in C(\mathbb{R}^d)} \frac{1}{n} \sum_{i=1}^n [h(f(x_i) - f(y_i))]. \tag{16}$$

Our analysis is about the geometry of $\phi_{\text{R}}(Y, X)$ in Eq. (16). In practical SGDA (stochastic GDA), at each iteration we draw a mini-batch of samples and update the parameters based on the mini-batch. The samples of true data $x_i$ are re-used multiple times (similar to SGD for a finite-sum optimization), but the samples of latent variables $z_i$ are fresh (similar to on-line optimization). Due to the re-use of true data, stochastic GDA shall be viewed as an online optimization algorithm for solving Eq. (16) where $x_i$'s can be the same. Recall that in the main results, we have assumed that $x_i$'s are distinct, thus there is a gap between our results and practice. Extending our results to the case of non-distinct $x_i$'s requires extra work. This was done in Claim C.1 for the 2-cluster setting. But for readability we do not further study this setting in the more general cases. We leave this to future work.

## G.3 Generalization and overfitting of GAN

One may wonder whether fitting the empirical distribution can cause memorization and failure to generate new data. Arora et al. [5] proved that for many GANs (including JS-GAN) with neural nets, only a polynomial number of samples are needed to achieve a small generalization error. We suspect that a similar generalization bound can be derived for RpGAN.

We provide some intuition why fitting the empirical data distribution via a GAN may avoid overfitting. Consider learning a two-cluster distribution as shown in Fig. 24. During training, we learn a generator that maps the latent samples $z_i$ to $x_i$, thus fitting the empirical distribution. If we sample a new latent sample $z_i$, then the generator will map $z_j$ to a new point $x_j$ in the underlying data distribution (due to the continuity of the generator function). Thus the continuity of the generator (or the restricted power of the generator) provides regularization for achieving generalization.

Figure 24: How to generate new point.

# H Proofs for Section 3 (2-Point Case) and Appendix C (2-Cluster Case)

We now provide the proofs for the toy results (i.e., the case $n = 2$).

## H.1 Proof of Claim 3.1 and Corollary 3.1 (for JS-GAN)

**Proof of Claim 3.1:** We will compute values of $\phi_{\text{JS}}(Y, X)$ for all $Y$. Recall $D$ can be any continuous function with range $(0, 1)$. Recall that $\phi_{\text{JS}}(Y, X) = \sup_D \frac{1}{2n} [\sum_{i=1}^n \log(D(x_i)) + \sum_{i=1}^n \log(1 - D(y_i))]$. Consider four cases. Denote a multiset $\mathcal{Y} = \{y_1, y_2\}$, and let $m_i = |\mathcal{Y} \cap \{x_i\}|, i \in \{1, 2\}$.

**Case 1** (state 1): $m_1 = m_2 = 1$. Then the objective is

$$\sup_D \frac{1}{2} \left[ \frac{1}{2} \log(D(x_1)) + \frac{1}{2} \log(1 - D(x_1)) + \frac{1}{2} \log(D(x_2)) + \frac{1}{2} \log(1 - D(x_2)) \right].$$

The optimal value is $-\log 2$, which is achieved when $D(x_1) = D(x_2) = \frac{1}{2}$.

**Case 2** (state 1a): $\{m_1, m_2\} = \{0, 1\}$. WLOG, assume $m_1 = 1, m_2 = 0$, and $y_1 = x_1, y_2 \notin \{x_1, x_2\}$. The objective becomes

$$\sup_D \frac{1}{2} \left[ \frac{1}{2} \log(D(x_1)) + \frac{1}{2} \log(D(x_2)) + \frac{1}{2} \log(1 - D(x_1)) + \frac{1}{2} \log(1 - D(y_2)) \right].$$

The optimal value $-\log 2/2$ is achieved when $D(x_1) = 1/2$, $D(x_2) \to 1$ and $D(y_2) \to 0$.

**Case 3** (state 1b): $\{m_1, m_2\} = \{0, 2\}$. WLOG, assume $y_1 = y_2 = x_1$. The objective becomes

$$\sup_D \frac{1}{2} \left[ \frac{1}{2} \log(D(x_1)) + \log(1 - D(x_1)) + \frac{1}{2} \log(D(x_2)) \right].$$

The optimal value $\frac{1}{4} \log \frac{1}{3} + \frac{1}{2} \log \frac{2}{3} \approx -0.4774$ is achieved when $D(x_1) = 1/3$ and $D(x_2) \to 1$.

**Case 4** (state 2): $m_1 = m_2 = 0$, i.e., $y_1, y_2 \notin \{x_1, x_2\}$. The objective is:

$$\sup_D \frac{1}{2} \left[ \frac{1}{2} \log(D(x_1)) + \frac{1}{2} \log(D(x_2)) + \frac{1}{2} \log(1 - D(y_1)) + \frac{1}{2} \log(1 - D(y_2)) \right].$$

These terms are independent, thus each term can achieve its supreme $\log 1 = 0$. Then the optimal value 0 is achieved when $D(x_1) = D(x_2) \to 1$ and $D(y_1) = D(y_2) \to 0$.

**Proof of Corollary 3.1**: Suppose $\epsilon$ is the minimal non-zero distance between two points of $x_1, x_2, y_1, y_2$. Consider a small perturbation of $\bar{Y}$ as $Y = (\bar{y_1} + \epsilon_1, \bar{y_2} + \epsilon_2)$, where $|\epsilon_i| < \epsilon$. We want to verify that

$$\phi(\bar{Y}, X) > \phi(Y, X) \approx -0.48. \tag{17}$$

There are two possibilities. **Possibility 1**: $\epsilon_1 = 0$ or $\epsilon_2 = 0$. WLOG, assume $\epsilon_1 = 0$, then we must have $\epsilon_2 > 0$. Then we still have $y_1 = \bar{y_1} = x_1$. Since the perturbation amount is small enough, we have $y_2 \notin \{x_1, x_2\}$. According to Case 2 above, we have $\phi(\bar{Y}, X) = -\log 2 \approx -0.35 > -0.48$. **Possibility 2**: $\epsilon_1 > 0, \epsilon_2 > 0$. Since the perturbation amount $\epsilon_1$ and $\epsilon_2$ are small enough, we have $y_1 \notin \{x_1, x_2\}, y_2 \notin \{x_1, x_2\}$. According to Case 4 above, we have $\phi(\bar{Y}, X) = 0 > -0.48$. Combining both cases, we have proved Eq. (17). □

## H.2   Proof of Claim 3.2 (for RS-GAN)

This is the result of RS-GAN for $n = 2$. WLOG, assume $x_1 = 0, x_2 = 1$. Denote $g_{\text{RS}}(Y) \triangleq \phi_{\text{RS}}(Y, X) = \sup_{f \in C(\mathbb{R}^d)} \frac{1}{2} \log \frac{1}{1 + \exp(f(0) - f(y_1))} + \frac{1}{2} \log \frac{1}{1 + \exp(f(1) - f(y_2))}$. Denote $m_i = |\{y_i\} \cap \{x_i\}|$, $i = 1, 2$; note this definition is different from JS-GAN in App. H.1. Consider three cases.

**Case 1**: $m_1 = m_2 = 1$. If $y_1 = 0, y_2 = 1$, then $g_{\text{RS}}(Y) = \frac{1}{2}[\log 0.5 + \log 0.5] = -\log 2 \approx -0.6937$. If $y_1 = 1, y_2 = 0$, then

$$g_{\text{RS}}(Y) = \sup_{f \in \mathcal{F}} \frac{1}{2} \log \frac{1}{1 + \exp(f(0) - f(1))} + \frac{1}{2} \log \frac{1}{1 + \exp(f(1) - f(0))}$$

$$= \sup_{t \in \mathbb{R}} \left[ \frac{1}{2} \log \frac{1}{1 + \exp(t)} + \frac{1}{2} \log \frac{1}{1 + \exp(-t)} \right] = -\log 2.$$

**Case 2**: $\{m_1, m_2\} = \{0, 1\}$. WLOG, assume $y_1 = 0, y_2 \neq 1$ (note that $y_2$ can be 0). Then

$$g_{\text{RS}}(Y) \geq \sup_{f \in \mathcal{F}} \frac{1}{2} \log \frac{1}{1 + \exp(f(0) - f(0))} + \frac{1}{2} \log \frac{1}{1 + \exp(f(1) - f(y_2))}$$

$$= -\frac{1}{2} \log 2 + \sup_{t \in \mathbb{R}} \frac{1}{2} \log \frac{1}{1 + \exp(t)} = -\frac{1}{2} \log 2 \approx -0.3466.$$

The value is achieved when $f(1) - f(y_2) \to -\infty$.

**Case 3**: $m_1 = m_2 = 0$. Then

$$g_{\text{RS}}(Y) \geq \sup_{f \in \mathcal{F}} \frac{1}{2} \log \frac{1}{1 + \exp(f(0) - f(y_1))} + \frac{1}{2} \log \frac{1}{1 + \exp(f(1) - f(y_2))}$$

$$= \sup_{t_1 \in \mathbb{R}, t_2 \in \mathbb{R}} \frac{1}{2} \log \frac{1}{1 + \exp(t_1)} + \frac{1}{2} \log \frac{1}{1 + \exp(t_2)} = 0.$$

The value is achieved when $f(1) - f(y_2) \to -\infty$ and $f(0) - f(y_2) \to -\infty$.

The global minimal value is $-\log 2$, and the only global minima are $\{y_1, y_2\} = \{x_1, x_2\}$. In addition, from any $Y$, it is easy to verify that there is a non-decreasing path from $Y$ to a global minimum.

### H.3 Proofs for 2-Cluster Data (Possibly Imbalanced)

**Proof of Claim C.1.** The proof is built on the proof of Claim 3.1 in Appendix H.1.

We first consider a special case $|\mathcal{X}_1 \cap Y|=m, |\mathcal{X}_2 \cap \mathcal{Y}|=0$. This means that $m$ generated points are in mode 1, and the rest are in neither modes. The loss value can be computed as follows:

$$\phi_{\text{JS}}(Y, X) = \frac{1}{2n}\left[\alpha n \log(\frac{\alpha n}{\alpha n + m}) + m \log(1 - \frac{\alpha n}{\alpha n + m})\right]$$
$$= \frac{\alpha}{2}\log(\alpha n) + \frac{m}{2n}\log m - \frac{\alpha n + m}{2n}\log(\alpha n + m)) = q_\alpha(m).$$

In general, if $|\mathcal{X}_1 \cap \mathcal{Y}|=m_1, |\mathcal{X}_2 \cap \mathcal{Y}|=m_2$, then $\phi_{\text{JS}}(Y, X)$ can be divided into three parts: the first part is the sum of the terms that contain $x_1$ (including $x_i$'s and $y_j$'s that are equal to $x_1$), the second part is the sum of the terms that contain $x_n$ (including $x_i$'s and $y_j$'s that are equal to $x_n$), and the third part is the sum of the terms that contain $y_j$'s that are not in $\{x_1, x_n\}$. Similar to Case 3 above, the value of the first part is $q_\alpha(m_1)$, and the value of the second part is $q_{1-\alpha}(m_2)$. Similar to the above special case, the value of the third part is 0. Therefore, the loss value is $\phi_{\text{JS}}(Y, X) = q_\alpha(m_1) + q_{1-\alpha}(m_2)$.

It is easy to show that $q_\alpha(m_1) + q_{1-\alpha}(m_2) \geq -\log 2$, and the equality is achieved iff $m_1 = n\alpha, m_2 = n(1-\alpha)$, i.e., $\mathcal{Y} = \mathcal{X}_1 \cup \mathcal{X}_2$. $\square$

**Proof sketch of Corollary C.1.** After a small enough perturbation, we must have $m_1 \triangleq |\mathcal{X}_2 \cap \mathcal{Y}| \leq n_1, m_2 \triangleq |\mathcal{X}_1 \cap \mathcal{Y}| \leq n_2$. Since $q_\alpha(m)$ and $q_{1-\alpha}(m)$ are strictly decreasing functions of $m$, we have

$$\phi(Y, X) = q_\alpha(m_1) + q_{1-\alpha}(m_2) \leq q_\alpha(n_1) + q_{1-\alpha}(n_2) = \phi(\hat{Y}, X).$$

The equality holds iff $(m_1, m_2) = (n_1, n_2)$, i.e., $Y = \hat{Y}$. This means that if $(n_1, n_2) \neq (n\alpha, n(1-\alpha))$, then $\hat{Y}$ is a sub-optimal strict local minimum. $\square$

We skip the detailed proof, since other parts are similar to the proof of Corollary 3.1.

## I Proof of Theorem 1 (Landscape of Separable-GAN)

Denote $F(D; Y) = \frac{1}{2n}\sum_{i=1}^{n}[h_1(f(x_i)) + h_2(-f(y_i))] \leq 0$ (since $h_i(t) \leq 0, i = 1, 2$ for any $t$).

**Step 1: Compute the value of $\phi(\cdot, X)$ for each $Y$.** For any $i$, denote $M_i = \{j : y_j = x_i\}, m_i = |M_i| \geq 0, i = 1, 2, \ldots, n$. Then $m_1 + \cdots + m_n = n$. Denote $\Omega = M_1 \cup M_2 \cdots \cup M_n$. Then

$$\phi(Y, X) = \frac{1}{2n}\sup_f \sum_{i=1}^{n}[h_1(f(x_i)) + h_2(-f(y_i))] = \frac{1}{2n}\sup_f \left(\sum_{i=1}^{n}[h_1(f_1(x_i)) + m_i h_2(-f(x_i))] + \sum_{j \notin \Omega}h_2(-f(y_i))\right)$$

$$\overset{(i)}{=} \frac{1}{2n}\left(\sum_{i=1}^{n}\sup_{t_i \in \mathbb{R}}[h_1(t_i) + m_i h_2(-t_i)] + |\Omega^c|\sup_{t \in \mathbb{R}}h_2(t)\right) \overset{(ii)}{=} \frac{1}{2n}\sum_{i=1}^{n}\xi(m_i) \tag{18a}$$

$$\overset{(iii)}{\geq} \frac{1}{2n}\sum_{i=1}^{n}m_i\xi(1) = \frac{1}{2}\xi(1).$$

Here (i) is because $f(y_j), j \in \Omega$ are independent of $h(x_i)$'s and thus can be any values; (ii) is by the definition $\xi(m) = \sup_t[h_1(t) + mh_2(-t)]$ and Assumption 4.1 that $\sup_t h_2(t) = 0$; (iii) is due to the convexity of $\xi$ (note that $\xi$ is the supreme of linear functions). Furthermore, if there is a certain $m_i > 1$, then $\xi(m_i) + (m_i - 1)\xi(0) = \xi(m_i) > m_i\xi(1)$ (according to Assumption 4.2), causing (iii) to become a strict inequality. Thus the equality in (iii) holds iff $m_i = 1, \forall i$, i.e., $\{y_1, \ldots, y_n\} = \{x_1, \ldots, x_n\}$. Therefore, we have proved that $\phi(Y, X)$ achieves the minimal value $\frac{1}{2}\xi(1)$ iff $\{y_1, \ldots, y_n\} = \{x_1, \ldots, x_n\}$.

**Step 2: Sufficient condition for strict local-min.** Next, we show that if $Y$ satisfies $m_1 + m_2 + \cdots + m_n = n$ then $Y$ is a strict local-min. Denote $\delta = \min_{k \neq l}\|x_k - x_l\|$. Consider a small perturbation of $Y$ as $\bar{Y} = (\bar{y}_1, \bar{y}_2, \ldots, \bar{y}_n) = (y_1 + \epsilon_1, y_2 + \epsilon_2, \ldots, y_n + \epsilon_n)$, where $\|\epsilon_j\| < \delta, \forall j$ and $\sum_j \|\epsilon_j\|^2 > 0$. We want to prove $\phi(\bar{Y}, X) > \phi(Y, X)$.

Denote $\bar{m}_i = |\{j : \bar{y}_j = x_i\}|, i = 1, 2, \ldots, n$. Consider an arbitrary $j$. Since $y_j \in \{x_1, \ldots, x_n\}$, there must be some $i$ such that $y_j = x_i$. Together with $\|\bar{y}_j - y_j\| = \|\epsilon_j\| < \delta = \min_{k \neq l}\|x_k - x_l\|$, we have $\bar{y}_j \notin (\{x_1, x_2, \ldots, x_n\}\backslash\{x_i\})$. In other words, the only possible point in $\{x_1, \ldots, x_n\}$ that can coincide with $\bar{y}_j$ is $x_i$, and this happens only when $\epsilon_j = 0$. This implies $\bar{m}_i \leq m_i, \forall i$. Since

we have assumed $\sum_j \|\epsilon_j\|^2 > 0$, for at least one $i$ we have $\bar{m}_i < m_i$. Together with Assumption 4.3 that $\xi(m)$ is a strictly decreasing function in $m \in [0, n]$, we have $\phi(\bar{Y}, X) = \frac{1}{n} \sum_{i=1}^{n} \xi(\bar{m}_i) > \frac{1}{n} \sum_{i=1}^{n} \xi(m_i) = \phi(Y, X)$.

**Step 3**: **Sub-optimal strict local-min.** Finally, if $Y$ satisfies that $m_1 + m_2 + \cdots + m_n = n$ and $m_k \geq 2$ for some $k$, then $\phi(Y, X) > \frac{1}{2}\xi(0)$. Thus $Y$ is a sub-optimal strict local minimum. **Q.E.D.**

**Remark 1**: $\xi(m)$ is convex (it is the supreme of linear functions), thus we always have $\xi(m) = \xi(m) + (m-1)\xi(0) \geq m\xi(1)$. Assump. 4.2 states that the inequality is strict, thus it is slightly stronger than the convexity of $\xi$. By Assump. 4.1, we also have $h_1(t) + (m+1)h_2(-t) \leq h_1(t) + mh_2(-t)$, thus $\xi(n) \leq \xi(n-1) \leq \cdots \leq \xi(0)$. Assumption 4.3 states that the inequalities are strict. This holds if the maximizer of $h_1(t) + mh_2(-t)$ does not coincide with the maximizer of $h_2(t)$. Intuitively, if $h(t)$ is "substantially different" from a constant function, then Assump. 4.2 and Assump. 4.3 hold.

**Remark 2**: The upper bound 0 in Assumption 4.1 is not essential, and can be relaxed to any finite numbers (change other two assumptions accordingly). We skip the details.

# J  Proof of Theorem 2 (Landscape of RpGAN)

This proof is the longest one in this paper. We will focus on a proof for the special case of RS-GAN. The proof for general RpGAN is quite similar, and presented in Appendix J.3. Recall $\phi_{\text{RS}}(Y, X) = \sup_f \frac{1}{n} \sum_{i=1}^{n} \log \frac{1}{1+\exp(f(y_i)-f(x_i))}$.

**Theorem J.1.** *(special case of Theorem 2 for RS-GAN) Suppose $x_1, x_2, \ldots, x_n \in \mathbb{R}^d$ are distinct. The global minimal value of $\phi_{\text{RS}}(Y, X)$ is $-\log 2$, which is achieved iff $\{x_1, \ldots, x_n\} = \{y_1, \ldots, y_n\}$. Furthermore, any point is global-min-reachable for the function.*

**Proof sketch.** We compute the value of $g(Y) = \phi_{\text{RS}}(Y, X)$ for any $Y$, using the following steps:

(i) We build a graph with vertices representing distinct values of $x_i, y_i$ and draw directed edges from $x_i$ to $y_i$. This graph can be decomposed into cycles and trees.

(ii) Each vertex in a cycle contributes $-\frac{1}{n} \log 2$ to the value $g(Y)$.

(iii) Each vertex in a tree contributes 0 to the value $g(Y)$.

(iv) The value $g(Y)$ equals $-\frac{1}{n} \log 2$ times the number of vertices in the cycles.

The outline of this section is as follows. In the first subsection, we analyze an example as warm-up. Next, we prove Theorem J.1. The proofs of some technical lemmas will be provided in the following subsections. Finally, in Appendix J.3 we present the proof for Theorem 2.

## J.1  Warm-up Example

We prove that if $\{y_1, y_2, \ldots, y_n\} = \{x_1, \ldots, x_n\}$, then $Y$ is a global minimum of $g(Y)$.

Suppose $y_i = x_{\sigma(i)}$, where $(\sigma(1), \sigma(2), \ldots, \sigma(n))$ is a permutation of $(1, 2, \ldots, n)$. We can divide $\{1, 2, \ldots, n\}$ into finitely many cycles $C_1, C_2, \ldots, C_K$, where each cycle $C_k = (c_k(1), c_k(2), \ldots, c_k(m_k))$ satisfies $c_k(j+1) = \sigma(c_k(j))$, $j \in \{1, 2, \ldots, m_k\}$. Here $c_k(m_k+1)$ is defined as $c_k(1)$. Now we calculate the value of $g(Y)$.

$$g(Y) = \sup_f \frac{1}{n} \sum_{i=1}^{n} \log \frac{1}{1+\exp(f(y_i)-f(x_i))} \overset{(i)}{=} -\inf_f \frac{1}{n} \sum_{k=1}^{K} \sum_{i \in C_k} \log\left(1 + \exp(f(y_i) - f(x_i))\right)$$

$$= -\inf_f \frac{1}{n} \sum_{k=1}^{K} \sum_{j=1}^{m_k} \log\left(1 + e^{f(x_{c_k(j+1)}) - f(x_{c_k(j)})}\right) \overset{(ii)}{=} -\frac{1}{n} \sum_{k=1}^{K} \inf_f \sum_{j=1}^{m_k} \log\left(1 + e^{f(x_{c_k(j+1)}) - f(x_{c_k(j)})}\right)$$

$$= -\frac{1}{n} \sum_{k=1}^{K} \inf_{t_1, t_2, \ldots, t_{m_k} \in \mathbb{R}} \left[\sum_{j=1}^{m_k-1} \log\left(1 + \exp(t_{j+1} - t_j)\right) + \log\left(1 + \exp(t_1 - t_{m_k})\right)\right]$$

$$\overset{(iii)}{=} -\frac{1}{n} \sum_{k=1}^{K} m_k \log(1 + \exp(0)) = -\log 2.$$

Here (i) is because $\{1, 2, \ldots, n\}$ is the combination of $C_1, \ldots, C_K$ and $i \in C_k$ means that $i = c_k(j)$ for some $j$. (ii) is because $C_k$'s are disjoint and $f$ can be any continuous function; more specifically, the choice of $\{f(x_i) : i \in C_k\}$ is independent of the choice of $\{f(x_i) : i \in C_l\}$ for any $k \neq l$, thus we can take the infimum over each cycle (i.e., put "inf" inside the sum over $k$). (iii) is because $\sum_{j=1}^{m-1} \log(1 + \exp(t_{j+1} - t_j)) + \log(1 + \exp(t_1 - t_m))$ is a convex function of $t_1, t_2, \ldots, t_m$ and the minimum is achieved at $t_1 = t_2 = \cdots = t_m = 0$.

## J.2   Proof of Theorem J.1

This proof is divided into three steps. In Step 1, we compute the value of $g(Y)$ if all $y_i \in \{x_1, \ldots, x_n\}$. This is the major step of the whole proof. In Step 2, we compute the value of $g(Y)$ for any $Y$. In Step 3, we show that there is a non-decreasing continuous path from $Y$ to a global minimum.

**Step 1: Compute $g(Y)$ that all $y_i \in \{x_1, \ldots, x_n\}$.** Define

$$R(X) = \{Y : y_i \in \{x_1, \ldots, x_n\}, \forall i\}. \tag{19}$$

**Step 1.1: Build a graph and decompose it.** We fix $Y \in R(X)$. We build a directed graph $G = (V, A)$ as follows. The set of vertices $V = \{1, 2, \ldots, n\}$ represent $x_1, x_2, \ldots, x_n$. A directed edge $(i, j) \in A$ if $y_i = x_j$. In this case, there is a term $\log(1 + \exp(f(x_j) - f(x_i)))$ in $g(Y)$. It is possible to have a self-loop $(i, i)$, which corresponds to the case $y_i = x_i$. By Eq. (19), we have

$$g(Y) = -\inf_f \frac{1}{n} \sum_{i=1}^n \log\left(1 + e^{f(y_i) - f(x_i)}\right) = -\inf_f \frac{1}{n} \sum_{(i,j) \in A} \log\left(1 + e^{f(x_j) - f(x_i)}\right). \tag{20}$$

Each $y_i$ corresponds to a unique $x_j$, thus the out-degree of $i$, denoted as outdegree$(i)$, must be exactly 1. The in-degree of each $i$, denoted as indegree$(i)$, can be any number in $\{0, 1, \ldots, n\}$.

We will show that the graph $G$ can be decomposed into the union of cycles and trees (see App. J.2.1 for its proof, and definitions of cycles and trees). A graphical illustration is given in Figure 25.

**Lemma 1.** *Suppose $G = (V, A)$ is a directed graph and outdegree$(v) = 1, \forall v \in V$. Then:*

*(a) There exist cycles $C_1, C_2, \ldots, C_K$ and subtrees $T_1, T_2, \ldots, T_M$ such that each edge $v \in A$ appears either in exactly one of the cycles or in exactly one of the subtrees.*

*(b) The root of each subtree $u_m$ is a vertex of a certain cycle $C_k$. In addition, each vertex of the graph appears in exactly one of the following sets: $V(C_1), \ldots, V(C_K), V(T_1)\backslash\{u_1\}, \ldots, V(T_M)\backslash\{u_M\}$.*

*(c) There is at least one cycle in the graph.*

| (a) Eg 1 for Lemma 1 | (b) Eg 2, with self-loop | (c) Example graph for general case |

Figure 25: The first two figures are two connected component of a graph representing the case $y_i \in \{x_1, \ldots, x_n\}, \forall i$. The first figure contains 10 vertices and 10 directed edges. It can be decomposed into a cycle $(1, 2, 3, 4)$ and two subtrees: one subtree consists of edge $(10, 4)$ and vertices 10, 4, and another consists of edges $(8, 7), (9, 7), (7, 5), (6, 5), (5, 1)$. The second figure has one cycle being a self-loop, and two trees attached to it. The third figure is an example graph of the case that some $y_i \notin \{x_1, \ldots, x_n\}$. In this example, $n = 8$ (so 8 edges), and all $y_i$'s are in $\{x_1, \ldots, x_n\}$ except $y_6, y_7$. The two edges $(6, 9)$ and $(6, 9)$ indicate the two terms $h(f(y_6) - f(x_6))$ and $h(f(y_7) - f(x_7))$ in $g(Y)$. They have the same head 9, thus $y_6 = y_7$. The vertice 9 has out-degree 0, indicating that $y_6 = y_7 \notin \{x_1, \ldots, x_n\}$. This figure can be decomposed into two cycles and three subtrees. Finally, adding a self-loop $(9, 9)$ will generate a graph where each edge has outdegree 1 (this is the reduction done in Step 2).

Denote $\xi(y_i, x_i) = \log\left(1 + e^{f(y_i) - f(x_i)}\right)$. According to Lemma 1, we have

$$-ng(Y) = \inf_f \sum_{i=1}^n \xi(y_i, x_i) \geq \inf_f \left[\sum_{k=1}^K \sum_{i \in V(C_k)} \xi(y_i, x_i)\right] \triangleq g_{\text{cyc}}. \tag{21}$$

**Step 1.2: Compute** $g_{\text{cyc}}$. We then compute $g_{\text{cyc}}$. Since $C_k$ is a cycle, we have $X_k \triangleq \{x_i : i \in C_k\} = \{y_i : i \in C_k\}$. Since $C_k$'s are disjoint, we have $X_k \cap X_l = \emptyset, \forall k \neq l$. This implies that $f(x_i), f(y_i)$ for $i$ in one cycle $C_k$ are independent of the values corresponding to other cycles. Then $g_{\text{cyc}}$ can be decomposed according to different cycles:

$$g_{\text{cyc}} = \inf_f \left[ \sum_{k=1}^{K} \sum_{i \in V(C_k)} \log\left(1 + \exp(f(y_i) - f(x_i))\right) \right] = \sum_{k=1}^{K} \inf_f \sum_{i \in V(C_k)} \log\left(1 + \exp(f(y_i) - f(x_i))\right).$$

Similar to Warm-up example 1, the infimum for each cycle is achieved when $f(x_i) = f(x_j), \forall i, j \in V(C_k)$. In addition,

$$g_{\text{cyc}} = -\log 2 \sum_{k=1}^{K} |V(C_k)|. \tag{22}$$

**Step 1.3: Compute** $g(Y)$. According to Eq. (21) and Eq. (22), we have

$$-ng(Y) \geq \sum_{k=1}^{K} |V(C_k)| \log 2. \tag{23}$$

Denote $F(Y; f) = -\frac{1}{n} \sum_{i=1}^{n} \log\left(1 + e^{f(y_i) - f(x_i)}\right)$, then $g(Y) = \inf_f F(Y; f)$. We claim that for any $\epsilon > 0$, there exists a continuous function $f$ such that

$$-nF(Y; f) < \sum_{k=1}^{K} |V(C_k)| \log 2 + \epsilon. \tag{24}$$

Let $N$ be a large positive number such that

$$n \log\left(1 + \exp(-N)\right)) < \epsilon. \tag{25}$$

Pick a continuous function $f$ as follows.

$$f(x_i) = \begin{cases} 0, & i \in \bigcup_{k=1}^{K} V(C_k), \\ N \cdot \text{depth}(i), & i \in \bigcup_{m=1}^{M} V(T_m). \end{cases} \tag{26}$$

Note that the root $u_m$ of a tree $T_m$ is also in a certain cycle $C_k$, thus the value $f(x_{u_m})$ is defined twice in Eq. (26), but in both definitions its value is 0, thus the definition of $f$ is valid. For any $i \in V(C_k)$, suppose $y_i = x_j$, then both $i, j \in V(C_k)$ which implies $f(y_i) - f(x_i) = f(x_j) - f(x_i) = 0$. For any $i \in V(T_m) \setminus \{u_m\}$, suppose $y_i = x_j$, then by the definition of the graph $(i, j)$ is a directed edge of the tree $T_m$, which means that $\text{depth}(i) = \text{depth}(j) + 1$. Thus $f(y_i) - f(x_i) = f(x_j) - f(x_i) = -N$. In summary, for the choice of $f$ in Eq. (26), we have

$$f(y_i) - f(x_i) = \begin{cases} 0, & i \in \bigcup_{k=1}^{K} V(C_k), \\ -N, & i \in \bigcup_{m=1}^{M} V(T_m). \end{cases} \tag{27}$$

Denote $p = \sum_{k=1}^{K} |V(C_k)| \log 2$. For the choice of $f$ in Eq. (26), we have

$$
\begin{aligned}
-nF(Y; f) &= \sum_{i=1}^{n} \log\left(1 + e^{f(y_i) - f(x_i)}\right) \\
&= \left[ \sum_{k=1}^{K} \sum_{i \in V(C_k)} \log\left(1 + e^{f(y_i) - f(x_i)}\right) + \sum_{m=1}^{M} \sum_{i \in V(T_m) \setminus \{u_m\}} \log\left(1 + e^{f(y_i) - f(x_i)}\right) \right] \\
&\overset{(27)}{=} \left[ \sum_{k=1}^{K} \sum_{i \in V(C_k)} \log\left(1 + e^{0}\right) + \sum_{m=1}^{M} \sum_{i \in V(T_m) \setminus \{u_m\}} \log\left(1 + e^{-N}\right) \right] \\
&= \sum_{k=1}^{K} |V(C_k)| \log 2 + \sum_{k=1}^{M} (|V(T_m)| - 1) \log\left(1 + e^{-N}\right) \leq p + n \log\left(1 + e^{-N}\right) \overset{(25)}{<} p + \epsilon.
\end{aligned}
\tag{28}
$$

This proves Eq. (24). Combining the two relations given in Eq. (24) and Eq. (23), we have

$$g(Y) = \inf_f F(Y; f) = \frac{1}{n} \sum_{k=1}^{K} |V(C_k)| \log 2, \ \forall \ Y \in R(X). \tag{29}$$

**Step 2: Compute $g(Y)$ for any $Y$.**

In the general case, not all $y_i$'s lie in $\{x_1, \ldots, x_n\}$. We will reduce to the previous case. Denote

$$H = \{i : y_i \in \{x_1, \ldots, x_n\}\}, \quad H^c = \{j : y_j \notin \{x_1, \ldots, x_n\}\}.$$

Since $y_j$'s in $H^c$ may be the same, we define the set of such distinct values of $y_j$'s as

$$Y_{\text{out}} = \{y \in \mathbb{R}^d : y = y_j, \text{ for some } j \in H^c\}.$$

Let $\bar{n} = |Y_{\text{out}}|$, then there are total $n + \bar{n}$ distinct values in $x_1, \ldots, x_n, y_1, \ldots, y_n$. WLOG, assume $y_1, \ldots, y_{\bar{n}}$ are distinct (this is because the value of $g(Y)$ does not change if we re-index $x_i$'s and $y_i$'s as long as the subscripts of $x_i, y_i$ change together), then

$$Y_{\text{out}} = \{y_1, \ldots, y_{\bar{n}}\}.$$

We create artificial "true data" and "fake data" $x_{n+1} = x_{n+1} = y_1, \ldots, x_{n+\bar{n}} = y_{n+\bar{n}} = y_{\bar{n}}$. Define $F_{\text{auc}}(Y, f) = -\sum_{i=1}^{n+m} \log \left(1 + e^{f(y_i) - f(x_i)}\right)$ $g_{\text{auc}} = -\inf_f F_{\text{auc}}(Y, f)$. Clearly, $F_{\text{auc}}(Y, f) = nF(Y, f) - \bar{n} \log 2$ and $ng(Y) = g_{\text{auc}} - \bar{n} \log 2$.

Consider the new configurations $\hat{X} = (x_1, \ldots, x_{n+\bar{n}})$ and $\hat{Y} = (y_1, \ldots, y_{n+\bar{n}})$. For the new configurations, we can build a graph $\hat{G}$ with $n + \bar{n}$ vertices and $n + \bar{n}$ edges. There are $K$ self-loops $C_{K+1}, \ldots, C_{K+\bar{n}}$ at the vertices corresponding to $y_1, \ldots, y_{\bar{n}}$. Based on Lemma 1, we have: (a) There exist cycles $C_1, C_2, \ldots, C_K, C_{K+1}, \ldots, C_{K+\bar{n}}$ and subtrees $T_1, T_2, \ldots, T_M$ (with roots $u_m$'s) s.t. each edge $v \in A$ appears in exactly one of the cycle or subtrees. (b) $u_m$ is a vertex of a certain cycle $C_k$ where $1 \le k \le K + \bar{n}$. (c) Each vertex of the graph appears in exactly one of the following sets: $V(C_1), \ldots, V(C_{K+\bar{n}}), V(T_1) \backslash \{u_1\}, \ldots, V(T_M) \backslash \{u_M\}$. According to the proof in Step 1, we have $g_{\text{auc}} = \sum_{k=1}^{K+\bar{n}} |V(C_k)| \log 2 = \sum_{k=1}^{K} |V(C_k)| \log 2 + \bar{n} \log 2$. Therefore,

$$ng(Y) = g_{\text{auc}} - \bar{n} \log 2 = \sum_{k=1}^{K} |V(C_k)| \log 2.$$

We build a graph $G$ by removing the self-loops $C_{K+j} = (y_j, y_j), j = 1, \ldots, \bar{n}$ in $\hat{G}$. The new graph $G$ consists of $n + \bar{n}$ vertices corresponding to $x_1, \ldots, x_n$ and $y_1, \ldots, y_{\bar{n}}$ and $n$ edges. The graph can be decomposed into cycles $C_1, C_2, \ldots, C_K$ (since $\bar{n}$ cycles are removed from $\hat{G}$) and subtrees $T_1, T_2, \ldots, T_M$. The value $ng(Y) = \sum_{k=1}^{K} |V(C_k)| \log 2$, where $C_k$'s are all the cycles of $G$.

**Step 3: Finding a non-decreasing path to a global minimum**. Finally, we prove that for any $Y$, there is a non-decreasing continuous path from $Y$ to one global minimal $Y^*$. The following claim shows that we can increase the value of $Y$ incrementally. See the proof in Appendix J.2.2.

**Claim J.1.** *For an arbitrary $Y$ that is not a global minimum, there exists another $\hat{Y}$ and a non-decreasing continuous path from $Y$ to $\hat{Y}$ such that $g(\hat{Y}) - g(Y) \ge \frac{1}{n} \log 2$.*

For any $Y$ that is not a global minimum, we apply Claim J.1 for finitely many times (no more than $n$ times), then we will arrive at one global minimum $Y^*$. We connect all non-decreasing continuous paths and get a non-decreasing continuous path from $Y$ to $Y^*$. This finishes the proof.

### J.2.1 Graph Preliminaries and Proof of Lemma 1

We present a few definitions from standard graph theory.

**Definition J.1.** *(walk, path, cycle) In a directed graph $G = (V, A)$, a walk $W = (v_0, e_1, v_1, e_2, \ldots, v_{m-1}, e_m, v_m)$ is a sequence of vertices and edges such that $v_i \in V, \forall \ i \in \{0, 1, \ldots, m\}$ and $e_i = (v_{i-1}, v_i) \in A, \forall \ i \in \{1, \ldots, m\}$. If $v_0, v_1, \ldots, v_m$ are distinct, we call it path (with length $m$). If $v_0, v_1, \ldots, v_{m-1}$ are distinct and $v_m = v_0$, we call it a cycle.*

Any $v$ has a path to itself (with length $0$), no matter whether there is an edge between $v$ to itself or not. This is because the degenerate walk $W = (v)$ satisfies the above definition. The set of vertices and edges in $W$ are denoted as $V(W)$ and $A(W)$ respectively.

**Definition J.2.** *(tree) A directed tree is a directed graph $T = (V, A)$ with a designated node $r \in V$, the root, such that there is exactly one path from $v$ to $r$ for each node $v \in V$ and there is no edge from the root $r$ to itself. The depth of a node is the length of the path from the node to the root (the depth of the root is $0$). A subtree of a directed graph $G$ is a subgraph $T$ which is a directed tree.*

**Proof of Lemma 1:**

We slightly extend the definition of "walk" to allow infinite length. We present two observations.

**Observation 1**: Starting from any vertex $v_0 \in V(G)$, there is a unique walk with infinite length

$$W(v_0) \triangleq (v_0, e_1, v_1, e_2, v_2, \ldots, v_i, e_i, v_{i+1}, e_{i+1}, \ldots),$$

where $e_i$ is an edge in $A(G)$ with tail $v_{i-1}$ and head $v_i$.

Proof of Observation 1: At each vertex $v_i$, there is a unique outgoing edge $e_i = (v_i, v_{i+1})$ which uniquely defines the next vertex $v_{i+1}$. Continue the process, we have proved Observation 1.

**Observation 2**: The walk $W(v_0) \triangleq (v_0, e_1, v_1, e_2, v_2, \ldots, v_i, e_i, v_{i+1}, e_{i+1}, \ldots)$ can be decomposed into two parts $W_1(v_0) = (v_0, e_1, v_1, e_2, v_2, \ldots, v_{i_0-1}, e_{i_0}, v_{i_0})$, $W_2(v_0) = (v_{i_0}, e_{i_0+1}, v_{i_0+1}, e_{i_0+2}, v_{i_0+2}, \ldots)$, where $W_1(v_0)$ is a path from $v_0$ to $v_{i_0}$ (i.e. $v_0, v_1, \ldots, v_{i_0}$ are distinct), and $W_2(v_0)$ is the repetition of a certain cycle (i.e., there exists $T$ such that $v_{i+T} = v_i$, for any $i \geq i_0$). This decomposition is unique, and we say the "first-touch-vertex" of $v_0$ is $v_{i_0}$.

**Proof of Observation 2**: Since the graph is finite, then some vertices must appear at least twice in $W(v_0)$. Among all such vertices, suppose $u$ is the one that appears the earliest in the walk $W(v_0)$, and the first two appearances are $v_{i_0} = u$ and $v_{i_1} = u$ and $i_0 < i_1$. Denote $T = i_1 - i_0$. Then it is easy to show $W_2(v_0)$ is the repetitions of the cycle consisting of vertices $v_{i_0}, v_{i_0+1}, \ldots, v_{i_1-1}$, and $W_1(v_0)$ is a directed path from $v_0$ to $v_{i_0}$.

The first-touch-vertex $u = v_{i_0}$ has the following properties: (i) $u \in C_k$ for some $k$; (ii) there exists a path from $v$ to $u$; (iii) any paths from $v$ to any vertex in the cycle $C_k$ other than $u$ must pass $u$. Note that if $u$ is in some cycle, then its first-touch-vertex is $u$ itself.

As a corollary of Observation 2, there is at least one cycle. Suppose all cycles of $G$ are $C_1, C_2, \ldots, C_K$. Because the outdegree of each vertex is $1$, these cycles must be disjoint, i.e., $V(C_i) \cap V(C_j) = \emptyset$ and $A(C_i) \cap A(C_j) = \emptyset$, for any $i \neq j$. Denote the set of vertices in the cycles as

$$V_c = \bigcup_{k=1}^{K} V(C_1) \cup \cdots \cup V(C_K). \tag{30}$$

Let $u_1, \ldots, u_M$ be the vertices of $C_1, \ldots, C_m$ with indegree at least $2$.

Based on Observation 2, starting from any vertex outside $V_c$ there is a unique path that reaches $V_c$. Combining all vertices that reach the cycles at $u_m$ (denoted as $V_m$), and the paths from these vertices to $u_m$, we obtain a directed subgraph $T_m$, which is connected with $V_c$ only via the vertex $u_m$. The subgraphs $T_m$'s are disjoint from each other since they are connected with $V_c$ via different vertices. In addition, each vertex outside of $V_c$ lies in exactly one of the subgraph $T_m$. Thus, we can partition the whole graph into the union of the cycles $C_1, \ldots, C_K$ and the subgraphs $T_1, \ldots, T_M$.

We then show $T_m$'s are trees. For any vertex $v_0$ in the subgraph $T_m$, consider the walk $W(v_0)$. Any path starting from $v_0$ must be part of $W(v_0)$. Starting from $v_0$ there is only one path from $v_0$ to $u_m$ which is $W_1(v_0)$, according to Observation 2. Therefore, by the definition of a directed tree, $T_m$ is a directed tree with the root $u_m$. Therefore, we can partition the whole graph into the union of the cycles $C_1, \ldots, C_K$ and subtrees $T_1, \ldots, T_M$ with disjoint edge sets; in addition, the edge sets of the cycles are disjoint, and the root of $T_l$ must be in certain cycle $C_k$. It is easy to verify the properties stated in Lemma 1. This finishes the proof.

### J.2.2 Proof of Claim J.1

We first prove the case for $d \geq 2$. Suppose the corresponding graph for $Y$ is $G$, and $G$ is decomposed into the union of cycles $C_1, \ldots, C_K$ and trees $T_1, \ldots, T_m$. We perform the following operation: pick

an arbitrary tree $T_m$ with the root $u_m$. The tree is non-empty, thus there must be an edge $e$ with the head $u_m$. Suppose $v$ is the tail of the edge $e$. Now we remove the edge $e = (v, u_m)$ and create a new edge $e' = (v, v)$. The new edge corresponds to $y_v = x_v$. The old edge $(v, u_m)$ corresponds to $y_v = x_{u_m}$ (and a term $h(f(x_{u_m}) - f(x_v))$) if $u_m \le n$ or $y_v = y_{u_m-n} \notin \{x_1, \dots, x_n\}$ (and a term $h(f(y_{u_m-n}) - f(x_v))$) if $u_m > n$. This change corresponds to the change of $y_v$: we change $y_v = x_{u_m}$ (if $u_m \le n$) or $y_v = y_{u_m-n}$ (if $u_m > n$) to $\hat{y}_v = x_v$. Let $\hat{y}_i = y_i$ for any $i \ne v$, and $\hat{Y} = (\hat{y}_1, \dots, \hat{y}_n)$ is the new point.

Previously $v$ is in a tree $T_m$ (not its root), now $v$ is the root of a new tree, and also part of the new cycle (self-loop) $C_{K+1} = (v, e', v)$. In this new graph, the number of vertices in cycles increases by 1, thus the value of $g$ increases by $-\frac{1}{n} \log 2$, i.e., $g(\hat{Y}) - g(Y) = \frac{1}{n} \log 2$.

Since $d \ge 2$, we can find a path in $\mathbb{R}^d$ from a point to another point without passing any of the points in $\{x_1, \dots, x_n\}$. In the continuous process of moving $y_v$ to $\hat{y}_v$, the function value will not change except at the end that $y_v = x_v$. Thus there is a non-increasing path from $Y$ to $\hat{Y}$, in the sense that along this path the function value of $g$ does not decrease.

The illustration of this proof is given below.

(a) Original graph

(b) Modified graph, with improved function value

Figure 26: Illustration of the proof of Claim J.1. For the figure on the left, we pick an arbitrary tree with the head being vertex 9, which corresponds to $y_6 = y_7$. We change $y_7$ to $\hat{y}_7 = x_7$ to obtain the figure on the right. Since one more cycle is created, the function value increases by $-\frac{1}{n} \log 2$.

For the case $d = 1$, the above proof does not work. The reason is that the path from $y_v$ to $\hat{y}_v$ may touch other points in $\{x_1, \dots, x_n\}$ and thus may change the value of $g$. We only need to make a small modification: we move $y_v$ in $\mathbb{R}$ until it touches a certain $x_i$ that corresponds to a vertex in the tree $T_m$, at which point a cycle is created, and the function value increases by at least $\frac{1}{n} \log 2$. This path is a non-decreasing path, thus the claim is also proved.

### J.3 Proof of Theorem 2

Obviously, $g(Y) \triangleq \phi_R(Y, X) = \frac{1}{n} \sup_{f \in C(\mathbb{R}^d)} \sum_{i=1}^n [h(f(x_i) - f(y_i))] \ge h(0)$ (by picking $f = 0$).

**Step 1: achieving optimal $g(Y)$.** We prove if $\{y_1, \dots, y_n\} = \{x_1, \dots, x_n\}$, then $g(Y) = h(0)$.

**Claim J.2.** *Assume $h$ is concave. Then the function $\xi_R(m) = \sup_{(t_1, \dots, t_k) \in ZO(m)} \sum_{i=1}^m h(t_i)$ satisfies $\xi_R(m) = mh(0)$, where the set $ZO(m) = \{t_1, t_2, \dots, t_m \in \mathbb{R} : \sum_{i=1}^m t_i = 0\}$.*

The proof of this claim is obvious and skipped here. When $\{y_1, \dots, y_n\} = \{x_1, \dots, x_n\}$, we can divide $[n]$ into multiple cycles $C_1 \cup \dots \cup C_K$, each with length $m_k$, and obtain $\phi_R(Y, X) = \frac{1}{n} \sup_{f \in C(\mathbb{R}^d)} \sum_{k=1}^K \sum_{i=1}^{m_k} [h(f(x_i) - f(y_i))] = \frac{1}{n} \sum_{k=1}^K \xi_R(m_k) = \frac{1}{n} \sum_{k=1}^K m_k h(0) = h(0)$.

**Step 2: compute $g(Y)$ when $y_i \in \{x_1, \dots, x_n\}, \forall i$.** Assume $y_i \in \{x_1, \dots, x_n\}, \forall i$. We build a directed graph $G = (V, A)$ as follows (the same graph as in Appendix J.2). The set of vertices $V = \{1, 2, \dots, n\}$ represents $x_1, x_2, \dots, x_n$. We draw a directed edge $(i, j) \in A$ if $y_i = x_j$. Note that it is possible to have a self-loop $(i, i)$, which corresponds to the case $y_i = x_i$.

According to Lemma 1, this graph can be decomposed into cycles $C_1, C_2, \dots, C_K$ and subtrees $T_1, T_2, \dots, T_M$. We claim that

$$\phi_R(Y, X) = \frac{1}{n} \sum_{k=1}^K |V(C_k)| h(0) \ge h(0). \tag{31}$$

The proof of the relation in Eq. (31) is similar to the proof of Eq. (22) used in the proof of Theorem 2, and briefly explained below. One major part of the proof is to show that the contribution of the nodes in the cycles is $\sum_{k=1}^{K} |V(C_k)| h(0)$. This is similar to Step 1, and is based on Claim J.2. Another major part of the proof is to show that the contribution of the nodes in the subtrees is zero, similar to the proof of Eq. (28). This is because we can utilize Assumption 4.4 to construct a sequence of $f$ values (similar to Eq. (26)) so that

$$ f(y_i) - f(x_i) = \begin{cases} 0, & i \in \bigcup_{k=1}^{K} V(C_k), \\ \alpha_N, & i \in \bigcup_{m=1}^{M} V(T_m). \end{cases} \tag{32} $$

Here $\{\alpha_N\}_{N=1}^{\infty}$ is a sequence of real numbers so that $\lim_{N \to \infty} h(\alpha_N) = \sup_t h(t) = 0$. In the case that $h(\infty) = 0$ like RS-GAN, we pick $\alpha_N = N$. In the case that $h(a) = 0$ for a certain finite number $a$, we can just pick $\alpha_N = a, \forall N$ (thus we do not need a sequence but just one choice).

Since the expression of $\phi_{\mathrm{R}}(Y, X)$ in Eq. (31) is a scaled version of the expression of $\phi_{\mathrm{RS}}(Y, X)$ (scale by $-\frac{\log 2}{h(0)}$), the rest of the proof is the same as the proof of Theorem 2.

**Step 3: function value for general $Y$ and GMR.** This step is the same as the proof of Theorem J.1. For the value of general $Y$, we build an "augmented graph" and apply the result in Step 2 to obtain $g(Y)$. To prove GMR, the same construction as the proof of Theorem J.1 suffices.

# K    Results in Parameter Space

We will first state the technical assumptions and then present the formal results in parameter space. The results become somewhat technical due to the complication of neural-nets. Suppose the discriminator neural net is $f_\theta$ where $\theta \in \mathbb{R}^J$ and the generator net is $G_w$ where $w \subset \mathbb{R}^K$.

**Assumption K.1.** *(representation power of discriminator net): For any distinct vectors $v_1, \ldots, v_{2n} \in \mathbb{R}^d$, any $b_1, \ldots, b_{2n} \in \mathbb{R}$, there exists $\theta \in \mathbb{R}^J$ such that $f_\theta(v_i) = b_i$, $i = 1, \ldots, 2n$.*

**Assumption K.2.** *(representation power of generator net in $\mathcal{W}$) For any distinct $z_1, \ldots, z_n \in \mathbb{R}^{d_z}$ and any $y_1, \ldots, y_n \in \mathbb{R}^d$, there exists $w \in \mathcal{W}$ such that $G_w(z_i) = y_i, i = 1, \ldots, n$.*

For any given $Z = (z_1, \ldots, z_n) \in \mathbb{R}^{d_z \times n}$, and any $\in \mathcal{W} \subseteq \mathbb{R}^K$, we define a set $G^{-1}(Y; Z)$ as follows: $w \in G^{-1}(Y; Z)$ iff $G_w(Z) = Y$ and $w \in \mathcal{W}$.

**Assumption K.3.** *(path-keeping property of generator net; duplication of Assumption 4.6): For any distinct $z_1, \ldots, z_n \in \mathbb{R}^{d_z}$, the following holds: for any continuous path $Y(t), t \in [0, 1]$ in the space $\mathbb{R}^{d \times n}$ and any $w_0 \in G^{-1}(Y(0); Z)$, there is continuous path $w(t), t \in [0, 1]$ such that $w(0) = w_0$ and $Y(t) = G_{w(t)}(Z), t \in [0, 1]$.*

We will present sufficient conditions for these assumptions later. Next we present two main results on the landscape of GANs in the parameter space.

**Proposition K.1.** *(formal version of Proposition 1) Consider the separable-GAN problem $\min_{w \in \mathbb{R}^K} \varphi_{\mathrm{sep}}(w)$, where $\varphi_{\mathrm{sep}}(w) = \sup_\theta \frac{1}{2n} \sum_{i=1}^{n} [h_1(f_\theta(x_i)) + h_2(-f_\theta(G_w(z_i)))]$ Suppose $h_1, h_2$ satisfy the same assumptions of Theorem 1. Suppose $G_w$ satisfies Assumption K.2 and Assumption 4.6 (with certain $\mathcal{W}$). Suppose $f_\theta$ satisfies Assumption K.1. Then there exist at least $(n^n - n!)$ distinct $w \in \mathcal{W}$ that are not global-min-reachable.*

**Proposition K.2.** *(formal version of Prop. 2) Consider the RpGAN problem $\min_{w \in \mathbb{R}^K} \varphi_{\mathrm{R}}(w)$, where $\varphi_{\mathrm{R}}(w) = \sup_\theta \frac{1}{n} \sum_{i=1}^{n} [h(f_\theta(x_i)) - f_\theta(G_w(z_i))]$. Suppose $h$ satisfies the same assumptions of Theorem 2. Suppose $G_w$ satisfies Assumption K.2 and Assumption 4.6 (with certain $\mathcal{W}$). Suppose $f_\theta$ satisfies Assumption K.1. Then any $w \in \mathcal{W}$ is global-min-reachable for $\varphi_{\mathrm{R}}(w)$.*

We have presented two generic results that relies on a few properties of the neural-nets. These properties can be satisfied by certain neural-nets, as discussed next. Our results largely rely on recent advanced in neural-net optimization theory.

## K.1    Sufficient Conditions for the Assumptions

In this part, we present a set of conditions on neural nets that ensure the assumptions to hold. We will discuss more conditions in the next subsection.

**Assumption K.4.** *(mildly wide) The last hidden layer has at least $\bar{n}$ neurons, where $\bar{n}$ is the number of input vectors.*

The assumption of width is common in recent theoretical works in neural net optimization (e.g. [50, 73, 2]). For the generator network, we set $\bar{n} = n$; for the discriminator network, we set $\bar{n} = 2n$.

**Assumption K.5.** *(smooth enough activation) The activation function $\sigma$ is an analytic function, and the $k$-th order derivatives $\sigma^{(k)}(0)$ are non-zero, for $k = 0, 1, 2, \ldots, \bar{n}$, where $\bar{n}$ is the number of input vectors.*

The assumption of the neuron activation is satisfied by sigmoid, tanh, SoftPlus, swish, etc.

For the generator network, consider a fully neural network $G_w(z) = W_H \sigma(W_{H-1} \ldots W_2 \sigma(W_1 z))$ that maps $z \in \mathbb{R}^{d_z}$ to $G_w(z) \in \mathbb{R}^d$. Define $T_k(z) = \sigma(W_{k-1} \ldots W_2 \sigma(W_1 z)) \in \mathbb{R}^{d_k}$ where $d_k$ is the number of neurons in the $k$-th hidden layer. Then we can write $G_w(z) = W_H T_H(z)$, where $W_H \in \mathbb{R}^{d \times d_H}$. Let $Z = (z_1, \ldots, z_n)$ and let $T_k(Z) = (T_k(z_1), \ldots, T_k(z_n)) \in \mathbb{R}^{d_k \times n}$, $k = 1, 2, \ldots, H$. Define $\mathcal{W} = \{w = (W_1, \ldots, W_H) : T_H(Z) \text{is full rank}\}$.

We will prove that under these two assumptions on the neural nets, the landscape of RpGAN is better than that of SepGAN.

**Proposition K.3.** *Suppose $h_1, h_2, h$ sastify assumptions in Theorem 1 and Theorem 2. Suppose $G_w, f_\theta$ satisfies Assump. K.5 and K.4 ($\bar{n} = n$ for $G_w$, and $\bar{n} = 2n$ for $f_\theta$). Then there exist at least $(n^n - n!)$ distinct $w \in \mathcal{W}$ that are not GMR for $\varphi_{\text{Sep}}(w)$. In contrast, any $w \in \mathcal{W}$ is global-min-reachable for $\varphi_{\text{R}}(w)$.*

This proposition is the corollary of Prop. K.1 and Prop. K.2; we only need to verify the assumptions in the two propositions. The following series of claims provide such verification.

**Claim K.1.** *Suppose Assumptions K.4 and K.5 hold for the generator net $G_w$ with distinct input $z_1, \ldots, z_n$. Then $\mathcal{W} = \{(W_1, \ldots, W_H) : T_H(Z) \text{ is full rank}\}$ is a dense set in $\mathbb{R}^K$. In addition, Assumption K.2 holds.*

This full-rank condition was used in a few works of neural-net landscape analysis (e.g. [72]). In GAN area, [7] studied invertible generator nets $G_w$ where the weights are restricted to a subset of $\mathbb{R}^K$ to avoid singularities. As the set $\mathcal{W}$ is dense, intuitively the iterates will stay in this set for most of the time. However, rigorously proving that the iterates stay in this set is not easy, and is one of the major challenges of current neural-network analysis. For instance, [38]) shows that for very wide neural networks with proper initialization along the training trajectory of gradient descent the neural-tangent kernel (a matrix related to $T_H(Z)$) is full rank. A similar analysis can prove that the matrix $T_H(Z)$ stays full rank during training under similar conditions. We do not attempt to develop the more complicated convergence analysis for general neural-nets here and leave it to future work.

**Claim K.2.** *Suppose Assumptions K.4 and K.5 hold for the generator net $G_w$ with distinct input $z_1, \ldots, z_n$. Then it satisfies Assumption 4.6 with $\mathcal{W}$ defined in Claim K.1.*

Assumption K.1 can be shown to hold under a similar condition to that in Claim K.1.

**Claim K.3.** *Consider a fully connected neural network $f_\theta(z) = \theta_H \sigma(\theta_{H-1} \ldots \theta_2 \sigma(\theta_1 z))$ that maps $u \in \mathbb{R}^d$ to $f_\theta(u) \in \mathbb{R}$ and suppose Assumptions K.4 and K.5 hold. Then Assumption K.1 holds.*

The proofs of the claims are given in Appendix K.5.

With these claims, we can immediately prove Prop. K.3.

**Proof of Prop. K.3.** According to Claim K.2, K.1, K.3, the assumptions of Prop. K.3 imply the assumptions of Prop. K.1 and Prop. K.2. Therefore, the conclusions of Prop. K.1 and Prop. K.2 hold. Since the conclusion of Prop. K.3 is the combination of the the conclusions of Prop. K.1 and Prop. K.2, it also holds. □

## K.2    Other Sufficient Conditions

Assumption K.3 (path-keeping property) is the key assumption. Various results in neural-net theory can ensure this assumption (or its variant) holds, and we have utilized one of the simplest such results in the last subsection. We recommend to check [80] which describes a bigger picture about various landscape results. In this subsection, we briefly discuss other possible results applicable to GAN.

We start with a strong conjecture about neural net landscape, which only requires a wide final hidden layer but no condition on the depth and activation.

**Conjecture K.1.** *Suppose $g_\theta$ is a fully connected neural net with any depth and any continuous activation, and it satisfies Assumption K.4 (i.e. a mildly wide final hidden layer). Assume $\ell(y, \hat{y})$ is convex in $\hat{y}$, then the empirical loss function of a supervised learning problem $\sum_{i=1}^n \ell(y_i, g_\theta(x_i))$ is global-min-reachable for any point.*

We then describe a related conjecture for GAN, which is easy to prove if Conjecture K.1 holds.

**Conjecture 1** (informal): Suppose $G_w$ is a fully connected net satisfying Assump. K.4 (i.e. a mildly wide final hidden layer). Suppose $G_w$ and $f_\theta$ are expressive enough (i.e. Assump. K.2 and Assump. K.1 hold). Then the RpGAN loss has a benign landscape, in the sense that any point is GMR for $\varphi_R(w)$. In contrast, the SepGAN loss does not have this property.

Unfortunately, we are not aware of any existing work that has proved Conjecture K.1, thus we are not able to prove Conjecture 1 above for now. Venturi et al. [84] proved a special case of Conjecture K.1 for $L = 1$ (one hidden layer), and other works such as Li et al. [50] prove a weaker version of Conjecture K.1; see [80] for other related results. The precise version of Conjecture K.1 seems non-trivial to prove.

We list two results on GAN that can be derived from weaker versions of Conjecture K.1; both results apply to the whole space instead of the dense subset $\mathcal{W}$.

**Result 1** (1-hidden-layer): Suppose $G_w$ is 1-hidden-layer network with any continuous activation. Suppose it satisfies Assump. K.4 (i.e. a mildly wide final hidden layer). Suppose $G_w$ and $f_\theta$ are expressive enough (i.e. Assump. K.2 and Assump. K.1 hold). Then the RpGAN loss satisfies GMR for any point. This result is based on Venturi et al. [84].

**Result 2**: Suppose $G_w$ is a fully connected network with any continuous activation and any number of layers. Suppose it satisfies Assump. K.4 (i.e. a mildly wide final hidden layer). Suppose $G_w$ and $f_\theta$ are expressive enough (i.e. Assump. K.2 and K.1 hold). Then the RpGAN loss has no sub-optimal set-wise local minima (see [50, Def. 1] for the definition). This result is based on Li et al. [50].

Due to space constraint, we do not present the proofs of the above two results (combining them with GANs is somewhat cumbersome). The high-level proof framework is similar to that of Prop. K.3.

## K.3 Proofs of Propositions for Parameter Space

**Proof of Proposition K.1.** The basic idea is to build a relation between the points in the parameter space to the points in the function space.

Denote $\mathcal{L}_{\text{sep}}(w; \theta) = \frac{1}{2n} \sum_{i=1}^n [h_1(f_\theta(x_i)) + h_2(-f_\theta(G_w(z_i)))]$, then $\varphi_{\text{sep}}(w) = \sup_\theta \mathcal{L}_{\text{sep}}(w; \theta)$. Denote $L_{\text{sep}}(Y; f) = \frac{1}{2n} \sum_{i=1}^n [h_1(f(x_i)) + h_2(-f(y_i))]$, and $\phi(Y, X) = \sup_f L_{\text{sep}}(Y; f)$. Note that in the definition of the two functions above, the discriminator is hidden in the sup operators, thus we have freedom to pick the discriminator values (unlike the generator space which we have to check all $w$ in the inverse of $Y$).

Our goal is to analyze the landscape of $\varphi_{\text{sep}}(w)$, based on the previously proved result on the landscape of $\phi(Y, X)$. We first show that the image of $\varphi_{\text{sep}}(\hat{w})$ is the same as that of $\phi_{\text{sep}}(\hat{Y}, X)$.

Define $G^{-1}(Y) \triangleq \{w : G_w(z_i) = y_i, i = 1, \ldots, n\}$. We first prove that

$$\phi_{\text{sep}}(\hat{Y}, X) = \varphi_{\text{sep}}(\hat{w}), \ \forall \ \hat{w} \in G^{-1}(\hat{Y}). \tag{33}$$

Suppose $\phi_{\text{sep}}(\hat{Y}, X) = \alpha$. This implies that $L_{\text{sep}}(\hat{Y}; f) \leq \alpha$ for any $f$; in addition, for any $\epsilon > 0$ there exists $\hat{f} \in C(\mathbb{R}^d)$ such that

$$L_{\text{sep}}(\hat{Y}; \hat{f}) \geq \alpha - \epsilon. \tag{34}$$

According to Assumption K.1, there exists $\theta^*$ such that $f_{\theta^*}(x_i) = \hat{f}(x_i), \ \forall \ i$, and $f_{\theta^*}(u) = \hat{f}(u), \forall \ u \in \{y_1, \ldots, y_n\} \backslash \{x_1, \ldots, x_n\}$. In other words, there exists $\theta^*$ such that

$$f_{\theta^*}(x_i) = \hat{f}(x_i), \ f_{\theta^*}(y_i) = \hat{f}(y_i), \ \forall \ i. \tag{35}$$

Then we have

$$\mathcal{L}_{\text{sep}}(\hat{w};\theta^*(\epsilon)) = \frac{1}{2n}\sum_{i=1}^n [h_1(f_{\theta^*}(x_i)) + h_2(-f_{\theta^*}(G_{\hat{w}}(z_i)))] \stackrel{(i)}{=} \sum_{i=1}^n [h_1(f_{\theta^*}(x_i)) + h_2(-f_{\theta^*}(\hat{y}_j))]$$

$$\stackrel{(ii)}{=} \frac{1}{2n}\sum_{i=1}^n [h_1(\hat{f}(x_i)) + h_2(-\hat{f}(\hat{y}_i))] = L_{\text{sep}}(\hat{Y};\hat{f}) \stackrel{(iii)}{\geq} \alpha - \epsilon.$$

In the above chain, (i) is due to the assumption $\hat{w} \in G^{-1}(\hat{Y})$ (which implies $G_{\hat{w}}(z_j) = \hat{y}_j$), (ii) is due to the choice of $\theta^*$. (iii) is due to (34).

Therefore, we have $\varphi_{\text{sep}}(\hat{w}) = \sup_\theta \mathcal{L}_{\text{sep}}(\hat{w};\theta) \geq \mathcal{L}_{\text{sep}}(\hat{w};\theta^*(\epsilon)) \geq \alpha - \epsilon$. Since this holds for any $\epsilon$, we have $\varphi_{\text{sep}}(\hat{w}) \geq \alpha$. Similarly, from $\mathcal{L}_{\text{sep}}(\hat{w};\theta) \leq \alpha$ we can obtain $\varphi_{\text{sep}}(\hat{w}) \leq \alpha$. Therefore $\varphi_{\text{sep}}(\hat{w}) = \alpha = \phi_{\text{sep}}(\hat{Y},X)$. This finishes the proof of (33).

Define

$$Q(X) \triangleq \{Y = (y_1,\ldots,y_n) \mid y_i \in \{x_1,\ldots,x_n\}, i \in \{1,2,\ldots,n\}; y_i = y_j \text{ for some } i \neq j\}.$$

Any $Y \in Q(X)$ is a mode-collapsed pattern. According to Theorem 1, any $Y \in Q(X)$ is a strict local minimum of $\phi_{\text{sep}}(Y,X)$, and thus $Y$ is not GMR. Therefore $\hat{w} \in G^{-1}(Y)$ where $Y \in Q(X)$ is not GMR; this is because a non-decreasing path in the parameter space will be mapped to a non-decreasing path in the function space, causing contradiction. Finally, according to Assumption K.2, for any $Y$ there exists at least one pre-image $w \in G^{-1}(Y) \cap \mathcal{W}$. There are $(n^n - n!)$ elements in $Q(X)$, thus there are at least $(n^n - n!)$ points in $\mathcal{W}$ that are not global-min-reachable. $\square$

**Proof of Proposition K.2.** Similar to Eq. (33), we have $\varphi_{\text{R}}(w) = \phi_{\text{R}}(Y,X)$ for any $w \in G^{-1}(Y)$. We need to prove that there is a non-decreasing path from any $w_0 \in \mathcal{W}$ to $w^*$, where $w^*$ is a certain global minimum. Let $Y_0 = G_{w_0}(z_1,\ldots,z_n)$. According to Thm. 2, there is a continuous path $Y(t)$ from $Y_0$ to $Y^*$ along which the loss value $\phi_{\text{R}}(Y(t),X)$ is non-increasing. According to Assump. 4.6, there is a continuous path $w(t)$ such that $w(0) = \hat{w}, Y(t) = G_{w(t)}(Z), t \in [0,1]$. Along this path, the value $\varphi_{\text{R}}(w(t)) = \phi_{\text{R}}(Y(t),X)$ is non-increasing, and at the end the function value $\varphi_{\text{R}}(w(1)) = \phi_{\text{R}}(Y^*,X)$ is the minimal value of $\varphi_{\text{R}}(w)$. Thus the existence of such a path is proved. $\square$

## K.4 A technical lemma

We present a technical lemma, that slightly generalizes [50, Proposition 1].

**Assumption K.6.** $v_1, v_2, \ldots, v_m \in \mathbb{R}^d$ are distinct, i.e., $v_i \neq v_j$ for any $i \neq j$.

**Lemma 2.** Define $T_H(V) = (\sigma(W_{H-1}\ldots W_2\sigma(W_1 v_i)))_{i=1}^m \in \mathbb{R}^{d_H \times m}$. Suppose Assumptions K.4, K.5 and K.6 hold. Then the set $\Omega = \{(W_1,\ldots,W_{H-1}) : \text{rank}(T_H(V)) < m\}$ has zero measure.

This claim is slightly different from [50, Proposition 1], which requires the input vectors to have one distinct dimension (i.e., there exists $j$ such that $v_{1j},\ldots,v_{m,j}$ are distinct); here we only require the input vectors to be distinct. It is not hard to link "distinct vectors" to "vectors with one distinct dimension" by a variable transformation.

**Claim K.4.** Suppose $v_1,\ldots,v_m \in \mathbb{R}^d$ are distinct. Then for generic matrix $W \in \mathbb{R}^{d \times d}$, for the vectors $\bar{v}_i = W v_i \in \mathbb{R}^d, i = 1,\ldots,n$, there exists $j$ such that $\bar{v}_{1j},\ldots,\bar{v}_{m,j}$ are distinct.

*Proof.* Define the set $\Omega_0 = \{u \mid u \in \mathbb{R}^{1 \times d}, \exists i \neq j \text{ s.t. } u^T v_i = u^T v_j\}$. This is the union of $d(d-1)$ hyperplanes $\Omega_{ij} \triangleq \{u \mid u \in \mathbb{R}^{1 \times d}, u^T v_i = u^T v_j\}$. Each hyperplane $\Omega_{ij}$ is a zero-measure set, thus the union of them $\Omega_0$ is also a zero-measure set. Let $u$ be the first row of $W$, then $u$ is generic vector and thus not in $\Omega_0$, which implies $\bar{v}_{11},\ldots,\bar{v}_{m,1}$ are distinct. $\square$

**Proof of Lemma 2:** Pick a generic matrix $A \in \mathbb{R}^{d_v \times d_v}$, then $\bar{v}_i = A v_i \in \mathbb{R}^{d_v \times 1}$ has one distinct dimension, i.e., there exists $j$ such that $\bar{v}_{1j},\ldots,\bar{v}_{m,j}$ are distinct. In addition, we can assume $A$ is full rank (since it is generic). Define

$$\bar{T}_H(\bar{V}) = (\sigma(W_{H-1}\ldots W_2\sigma(\bar{W}_1\bar{v}_1)),\ldots,\sigma(W_{H-1}\ldots W_2\sigma(\bar{W}_1\bar{v}_m)) \in \mathbb{R}^{d_H \times m}.$$

According to [50, Prop. 1], the set $\bar{\Omega} = \{(\bar{W}_1, W_2, W_3, \ldots, W_{H-1}) : \text{rank}(\bar{T}_H(\bar{V})) < m\}$ has zero measure. With the transformation $\eta_0(\bar{W}_1) = \bar{W}_1 A^{-1}$, we have $\sigma(W_{H-1}\ldots W_2\sigma(\bar{W}_1\bar{v}_i)) =$

$\sigma(W_{H-1}\ldots W_2\sigma(W_1 v_i))$, $\forall \; i$ and thus $\bar{T}_H(\bar{V}) = T_H(V)$. Define $\eta(\bar{W}_1, W_2, \ldots, W_m) = (\bar{W}_1 A^{-1}, W_2, \ldots, W_m)$, then $\eta$ is a homeomorphism between $\bar{\Omega}$ and $\Omega$. Therefore the set $\Omega = \{(W_1, \ldots, W_{H-1}) : \text{rank}(T_H(V)) < m\}$ has zero measure. $\square$

### K.5 Proof of claims

**Proof of Claim K.1:** According to Lemma 2, $\mathcal{W}$ is a dense subset of $\mathbb{R}^J$ (in fact, $\Omega$ is defined for a general neural network, and $\mathcal{W}$ is defined for the generator network, thus an instance of $\Omega$). As a result, there exists $(W_1, \ldots, W_{H-1})$ such that $T_H(Z)$ has rank at least $n$. Thus for any $y_1, y_2, \ldots, y_n \in \mathbb{R}^d$, there exists $W_H$ such that $W_H T_H(Z) = (y_1, \ldots, y_n)$. $\square$

**Proof of Claim K.2:** For any continuous path $Y(t), t \in [0, 1]$ in the space $\mathbb{R}^{d \times n}$, any $w_0 \in G^{-1}(Y(0))$ and any $\epsilon > 0$, our goal is to show that there exists a continuous path $w(t), t \in [0, 1]$ such that $w(0) = w_0$ and $Y(t) = G_{w(t)}(Z), t \in [0, 1]$.

Due to the assumption of $w_0 \in \mathcal{W}$, we know that $w_0$ corresponds to a rank-$n$ post-activation matrix $T_H(Z)$. Suppose $w_0 = (W_1, \ldots, W_H)$ and $T_H(Z) = (T_H(z_1), \ldots, T_H(z_n)) \in \mathbb{R}^{d_H \times n}$ has rank $n$. Since $T_H(Z)$ is full rank, for any path from $Y(0)$ to $Y(1)$, we can continuously change $W_H$ such that the output of $G_w(Z)$ changes from $Y(0)$ to $Y(1)$. Thus there exists a continuous path $w(t), t \in [0, 1]$ such that $w(0) = w_0$ and $Y(t) = G_{w(t)}(Z), t \in [0, 1]$. $\square$

**Proof of Claim K.3:** This is a direct application of Lemma 2. Different from Claim K.2, here we apply Lemma 2 to the discriminator network. $\square$

## L  Discussion of Wasserstein GAN

W-GAN is a popular formulation of GAN, so a natural question is whether we can prove a similar landscape result for W-GAN. Consider W-GAN formulation (empirical version) $\min_Y \phi_W(Y, X)$, where

$$\phi_W(Y, X) = \max_{|f|_L \leq 1} \frac{1}{n} \sum_{i=1}^{n} [f(x_i) - f(y_i)].$$

For simplicity we consider the same number of generated samples and true samples. It can be viewed as a special case of RpGAN where $h(t) = -t$; it can also be viewed as a special case of SepGAN where $h_1(t) = h_2(t) = -t$.

However, the major complication is the Lipschitz constraint. It makes the computation of the function values much harder. For the case of $n = 2$, the function value of $\phi_W(Y, X)$ is provided in the following claim.

**Claim L.1.** *Suppose $n = 2$. Denote $a_1 = x_1, a_2 = x_2, a_3 = y_1, a_4 = y_2$. The value of $\phi_W(Y, X)$ is*

$$\max_{u_1, u_2, u_3, u_4 \in \mathbb{R}} u_1 + u_2 - u_3 - u_4,$$
$$s.t. \quad |u_i - u_j| \leq \|a_i - a_j\|, \forall i, j \in \{1, 2, 3, 4\}.$$

This claim is not hard to prove, and we skip the proof here.

This claim indicates that computing $\phi_W(Y, X)$ is equivalent to solving a linear program (LP). Solving LP itself is computationally feasible, but our landscape analysis requires to infer about the global landscape of $\phi_W(Y, X)$ as a function of $Y$. In classical optimization, it is possible to state that the optimal value of an LP is a convex function of a certain parameter (e.g. the coefficient of the objective). But in our LP $y_i$'s appear in multiple positions of the LP, and we are not aware of an existing result that can be readily applied.

Similar to Kantorovich-Rubinstein Duality, we can write down the dual problem of the LP where the objective is linear combination of $\|a_i - a_j\|$. However, it is still not clear what to say about the global landscape, due to the lack of closed-form solutions.

Finally, we remark that although W-GAN has a strong theoretical appeal, it did not replace JS-GAN or simple variants of JS-GAN in recent GAN models. For instance, SN-GAN [67] and BigGAN [18] use hinge-GAN.

| (a) Generator | (b) Discriminator |
|---|---|
| $z \in \mathbb{R}^{128} \sim \mathcal{N}(0, I)$ | image $x \in [-1, 1]^{H \times W \times 3}$ |
| $128 \to h \times w \times 512/c$, dense, linear | $3 \times 3$, stride 1 conv, 64/c |
| $4 \times 4$, stride 2 deconv, 256/c, BN, ReLU | $4 \times 4$, stride 2 conv, 128/c<br>$3 \times 3$, stride 1 conv, 128/c |
| $4 \times 4$, stride 2 deconv, 128/c, BN, ReLU | $4 \times 4$, stride 2 conv, 256/c<br>$3 \times 3$, stride 1 conv, 256/c |
| $4 \times 4$, stride 2 deconv, 64/c, BN, ReLU | $4 \times 4$, stride 2 conv, 512/c<br>$3 \times 3$, stride 1 conv, 512/c |
| $3 \times 3$, stride 1 conv, 3, Tanh | $h \times w \times 512/c \to s$, linear |

Table 7: CNN models for CIFAR-10 and STL-10 used in our experiments on image Generation. h = w = 4, H = W = 32 for CIFAR-10. h = w = 6, H = W = 48 for STL-10. c=1, 2 and 4 for the regular, 1/2 and 1/4 channel structures respectively. All layers of D use LReLU-0.1 (except the final dense ''linear'' layer).

| (a) Generator | (b) Discriminator |
|---|---|
| $z \in \mathbb{R}^{128} \sim \mathcal{N}(0, I)$ | $x \in [-1, 1]^{256 \times 256 \times 3}$ |
| reshape $\to 128 \times 1 \times 1$ | $4 \times 4$, stride 2 conv, 32, |
| $4 \times 4$, stride 1 deconv, BN, 1024 | $4 \times 4$, stride 2 conv, 64 |
| $4 \times 4$, stride 2 deconv, BN, 512 | $4 \times 4$, stride 2 conv, 128 |
| $4 \times 4$, stride 2 deconv, BN, 256 | $4 \times 4$, stride 2 conv, 256 |
| $4 \times 4$, stride 2 deconv, BN, 128 | $4 \times 4$, stride 2 conv, 512 |
| $4 \times 4$, stride 2 deconv, BN, 64 | $4 \times 4$, stride 2 conv, 1024 |
| $4 \times 4$, stride 2 deconv, BN, 32 | dense $\to 1$ |
| $4 \times 4$, stride 2 deconv, 3, Tanh | |

Table 8: CNN model architecture for size 256 LSUN used in our experiments on high resolution image generation. All layers of G use ReLU (except one layer with Tanh); all layers of D use LReLU-0.1.

| (a) Generator | (b) Discriminator |
|---|---|
| $z \in \mathbb{R}^{128} \sim \mathcal{N}(0, I)$ | image $x \in [-1, 1]^{32 \times 32 \times 3}$ |
| dense, $4 \times 4 \times 256/c$ | ResBlock down 128/c |
| ResBlock up 256/c | ResBlock down 128/c |
| ResBlock up 256/c | ResBlock down 128/c |
| ResBlock up 256/c | ResBlock down 128/c |
| BN, ReLU, $3 \times 3$ conv, 3 Tanh | LReLU 0.1 |
| | Global sum pooling |
| | dense $\to 1$ |

Table 9: Resnet architecture for CIFAR-10. c=1, 2 and 4 for the regular, 1/2 and 1/4 channel structures respectively.

| (a) Generator | (b) Discriminator |
|---|---|
| $z \in \mathbb{R}^{128} \sim \mathcal{N}(0, I)$ | image $x \in [-1, 1]^{48 \times 48 \times 3}$ |
| dense, $6 \times 6 \times 512/c$ | ResBlock down 64/c |
| ResBlock up 256/c | ResBlock down 128/c |
| ResBlock up 128/c | ResBlock down 256/c |
| ResBlock up 64/c | ResBlock down 512/c |
| BN, ReLU, $3 \times 3$ conv, 3 Tanh | ResBlock down 1024/c |
| | LReLU 0.1 |
| | Global sum pooling |
| | dense $\to 1$ |

Table 10: Resnet architecture for STL-10. c=1, 2 and 4 for the regular, 1/2 and 1/4 channel structures respectively.

| (a) Generator | (b) Discriminator |
|---|---|
| $z \in \mathbb{R}^{128} \sim \mathcal{N}(0, I)$ | image $x \in [-1, 1]^{32 \times 32 \times 3}$ |
| dense, $4 \times 4 \times 128$ | BRes down (64, 32, 64) |
| BRes up (128, 64, 128) | BRes down (64, 32, 64) |
| BRes up (128, 64, 128) | BRes down (64, 32, 64) |
| BRes up (128, 64, 128) | BRes down (64, 32, 64) |
| BN, ReLU, $3 \times 3$ conv, 3 Tanh | LReLU 0.1 |
| | Global sum pooling |
| | dense $\to 1$ |

Table 11: BottleNeck Resnet models for CIFAR-10. BRes refers to BottleNeck ResBlock. BRes $(a, b, c)$ refers to the Bottleneck resblock with (input, hidden and output) being $(a, b, c)$.

| (a) Generator | (b) Discriminator |
|---|---|
| $z \in \mathbb{R}^{128} \sim \mathcal{N}(0, I)$ | image $x \in [-1, 1]^{48 \times 48 \times 3}$ |
| dense, $6 \times 6 \times 256$ | BRes down (3, 16, 32) |
| BRes up (256, 64, 128) | BRes down (32, 16, 64) |
| BRes up (128, 32, 64) | BRes down (64, 32, 128) |
| BRes up (64, 16, 32) | BRes down (128, 64, 256) |
| BN, ReLU, $3 \times 3$ conv, 3 Tanh | BRes down (256, 128, 512) |
| | LReLU 0.1 |
| | Global sum pooling |
| | dense $\to 1$ |

Table 12: BottleNeck Resnet models for STL-10.

| | **RS-GAN generator learning rate** | | |
|---|---|---|---|
| | | CIFAR-10 | STL-10 |
| CNN | No normalization | 2e-4 | 5e-4 |
| | Regular + SN | 5e-4 | 5e-4 |
| | channel/2 + SN | 5e-4 | 5e-4 |
| | channel/4 + SN | 2e-4 | 5e-4 |
| ResNet | Regular+SN | 1.5e-3 | 1e-3 |
| | channel/2 + SN | 1.5e-3 | 1e-3 |
| | channel/4 + SN | 1e-3 | 5e-4 |
| | BottleNeck | 1e-3 | 1e-3 |

| **WGAN-GP Hyper-parameters** | |
|---|---|
| generator learning rate | 1e-4 |
| discriminator learning rate | 1e-4 |
| $\beta_1$ | 0.5 |
| $\beta_2$ | 0.9 |
| Gradient penalty $\lambda$ | 10 |
| # D iterations per G iteration | 5 |

Table 13: Learning rate for RS-GAN in each setting. Hyper-parameters used for WGAN-GP

(a) real data

(b) JS-GAN + BatchNorm

(c) WGAN-GP

(d) RS-GAN

(e) JS-GAN + Spectral Norm + Regular CNN

(f) RS-GAN + Spectral Norm + Regular CNN

(g) JS-GAN + Spectral Norm + Channel/2

(h) RS-GAN + Spectral Norm + Channel/2

(i) JS-GAN + Spectral Norm + Channel/4

(j) RS-GAN + Spectral Norm + Channel/4

Figure 16: Generated CIFAR-10 samples with CNN.

(a) JS-GAN + Spectral Norm + Regular ResNet

(b) RS-GAN + Spectral Norm + Regular ResNet

(c) JS-GAN + Spectral Norm + Channel/2

(d) RS-GAN + Spectral Norm + Channel/2

(e) JS-GAN + Spectral Norm + Channel/4

(f) RS-GAN + Spectral Norm + Channel/4

(g) JS-GAN + Spectral Norm + BottleNeck

(h) RS-GAN + Spectral Norm + BottleNeck

Figure 17: Generated CIFAR-10 samples on ResNet.

(a) real data

(b) JS-GAN + BatchNorm

(c) WGAN-GP

(d) RS-GAN

(e) JS-GAN + Spectral Norm + Regular CNN

(f) RS-GAN + Spectral Norm + Regular CNN

(g) JS-GAN + Spectral Norm + Channel/2

(h) RS-GAN + Spectral Norm + Channel/2

(i) JS-GAN + Spectral Norm + Channel/4

(j) RS-GAN + Spectral Norm + Channel/4

Figure 18: Generated STL-10 samples with CNN.

(a) JS-GAN + Spectral Norm + Regular ResNet

(b) RS-GAN + Spectral Norm + Regular ResNet

(c) JS-GAN + Spectral Norm + Channel/2

(d) RS-GAN + Spectral Norm + Channel/2

(e) JS-GAN + Spectral Norm + Channel/4

(f) RS-GAN + Spectral Norm + Channel/4

(g) JS-GAN + Spectral Norm + BottleNeck

(h) RS-GAN + Spectral Norm + BottleNeck

Figure 19: Generated STL-10 samples with ResNet.

(a) LSUN Church by JS-GAN

(b) LSUN Tower by JS-GAN

(c) LSUN Church by RS-GAN

(d) LSUN Tower by RS-GAN

Figure 20: Generated $256 \times 256$ Church and Tower Image by JS-GAN and RS-GAN.

## Footnotes

[6]Note that max-sliced-WGAN in Deshpande et al. [25] uses $\min_Y \max_{\|v\| \le 1, |g|_L \le 1} W_2(v^T g(X), v^T g(Y))^2$, while sliced-WGAN in Deshpande et al. [24] uses $\min_Y \mathbb{E}_{\|v\|=1} \max_{|g|_L \le 1} W_2(v^T g(X), v^T g(Y))^2$. In Eq. (10) we use $f(u)$ to replace $v^T g(u)$ to simplify the expression; although technically, $f$ and $v^T g$ are not equivalent, this minor difference does not affect our discussion.

[7]Technically since we are not dealing with a pure minimization problem, we should say "the algorithm converges to a bad attractor". But for simplicity of illustration, we still call it "local minimum."

[8]We tuned glr in the set 2e-4, 5e-4, 1e-3, 1.5e-3 and find that glr = 2e-4 performs the best in most cases for SepGAN, so we follow the suggestion of [67, 76].