[Reviews · NeurIPS 2020]

Review 1

Summary and Contributions: Studying the global optimization landscape is a very important problem for understanding the training process of GANs. This work presents the global landscape of the original GAN and the RS-GAN. And theoretically prove that the RS-GAN has a better landscape and this better landscape can lead to better performance.

Strengths: 1. Clear analysis of the global landscape of the original GAN and point out the bad basin in the global landscape of the original GAN, which may lead to bad performance and mode collapse problem of GANs. 2. Based on the analysis of the global landscape of the original GAN, this paper improves the global landscape using the previously proposed RS-GAN. And prove that the global landscape of the RS-GAN has no bad local minima. The analysis of the 2-Point case is very clear to compare the landscape between JS-GAN and RS-GAN.

Weaknesses: 1. More analysis of the global optimization landscape should be conducted on various GANs like LSGAN, WGAN, and so on. This will make readers to fully understand the differences among different loss functions applied to GANs. 2. The experiments are not sufficient to validate the effectiveness of the proposed RS-GAN. This mainly lies in two aspects. First, some previous works also focused on the convergence and the design of the loss function of the GANs, this work should compare the results with them. Second, the results are marginally better than the simple baseline JS-GAN in terms of the Inception Score and FID on the CIFAR-10 dataset.

Correctness: Yes.

Clarity: Yes.

Relation to Prior Work: Yes, it is. But there can be more discussions on global landscape analysis of previous works.

Reproducibility: Yes

Additional Feedback:


Review 2

Summary and Contributions: This work studies the global landscape of GANs: (1) they showed the empirical versions of JS-GAN and prove that it has exponentially many bad basins, which are mode-collapse patterns; (2) they proved that a simple coupling idea (resulting in RS-GAN) can remove the bad basins in theory; (3) the landscape theory is verified based on a few experiments, e.g., RS-GAN has a bigger advantage over JS-GAN for narrow nets.

Strengths: This paper is well organized and well written overall, easy to follow. This work provides several interesting theoretical results on understanding GAN, and verified them experimentally. it suggests that mode-collapse patterns cause exponentially many bad local minimizers for original JS-GAN; It suggests a new way to improve the optimization landscape via a simple coupling idea (RS-GAN), which is proven to remove bad basins. The theoretical results are corroborated by experimental results on both synthetic and real datasets. Although there are still quite a lot of limitations, the overall results are nice and interesting.

Weaknesses: From the reviewer’s understanding, most of the theoretical understanding is for the unparameterized case. When the problem is parameterized with network structure (theta, w), the result showed here is still very vague. The connections need to be discussed in more details. Also, the existence of a reachable path does not necessarily mean that sgd can follow the path and non-aymptotically reach the solution, that there are still quite a lot questions to be addressed.

Correctness: Although the reviewer has now checked in detail about every proofs (55 pages) due to time and energy limitations, the reviewer believe the result is intuitive and sound.

Clarity: The paper is well-organized, but the reviewer finds the presentation can be improved especially in Section 2 and Section 3, with more high-level ideas and less details.

Relation to Prior Work: The literature review is very comprehensive

Reproducibility: Yes

Additional Feedback: 1. In Proposition 1, can the author provide more details in the main paper about the assumptions on the network? What does having enough representation power means? 2. In Proposition 2, it only suggests that w is reachable. Can the author claim similar things for theta? =============== After reading all reviewers' comments, and authors' feedback. Most of my concerns are addressed in the rebuttal, and I maintain my recommendation. This work provides several interesting theoretical results on understanding GAN through studying the landscape via mode-collapse patterns, which has led to a new way to improve the optimization landscape via a simple coupling idea (RS-GAN), which is proven to remove bad basins. The experimental results are convincing.


Review 3

Summary and Contributions: This paper studied the global optimization landscape of generative adversarial networks. First, the authors proved that the outer-minimization problem of JS-GAN has exponentially many bad strict local min, each of which corresponds to a mode-collapse situation. Then, they showed that RS-GAN that couples true data and generated data in the loss function has no bad basin. Finally, they verified the theoretical predictions in synthetic experiments and real data. In particular, they showed that RS-GAN performs better than JS-GAN for narrow nets. ---------------------------------------------------------------------------------------------------------- Thanks for the authors' response, which has addressed most of my concerns. This is a good theory paper that studied the optimization landscape of GANs, and I decided to keep my positive review.

Strengths: This is a good theory paper that studied the optimization landscape of GANs. They showed JS-GAN has many bad strict local min while RS-GAN has no bad basin. I think this paper is a good step towards understanding the optimization of GANs and also developing new GANs that are easier to train.

Weaknesses: 1. As a mostly theoretical paper, it might be good to include a proof sketch for Theorem 1 and Theorem 2 in the main text. In the current version, it is hard to get a sense of the proof without reading the supplementary material. 2. The landscape results in parameter space looks very surprising because it has no assumptions on the generator and discriminator architecture except for enough representation. This looks surprising to me because usually these kind of global optimization result for neural networks needs strong assumptions on the architecture. It would be good if the authors can give some intuition on the proof of Proposition 1,2 and also explain how the proof strategy differ from that for the optimization landscape of neural networks (in the supervised learning setting).

Correctness: Yes.

Clarity: Yes.

Relation to Prior Work: Yes.

Reproducibility: Yes

Additional Feedback:


Review 4

Summary and Contributions: Authors study the impact of the loss function on the convergence of Generative Adversarial Networks. In particular, they ask the following question: given two different objectives (JSGAN and RSGAN), which one does provide "the better" landscape ?

Strengths: 1) The paper is nicely written and clear. 2) Interestingly, this study aims at understanding the existence of mode collapse in the original formulation of GANs by Goodfellow et al. (2014). Their work shows that the JSGAN objective has many sub-optimal strict local minima (increasing with the number of distinct data points). This is, in particular, well illustrated in Figure 3 and Corollary 2.1. 3) Besides, the authors study the RSGAN formulation and show that the only basin is the global minimum and thus the GAN cannot get stuck in a "bad local minimum". Theorem 1 and 2 are extensions to the finite-sample case: they are clear, understandable and highlight the fact that the original formulation might lead to "bad" areas in the function space.

Weaknesses: 1)It would have been really interesting to study the WGAN objective, already known to solve most of the mode collapse problem. 2) If this works sheds light on the original formulation of GANs, it would also be insightful to understand what could be changed in this original objective to remove the local bad basins. In the paper "On convergence and stability of GANs", the authors already suggested that the mode collapse in the original formulation of GANs might come from the existence of multiple equilibria (and thus "bad ones") and proposed a regularization scheme (called DRAGAN) to solve the problem. It would have been interesting to make the link with this work and study the DRAGAN architecture. 3) Section 3 (two-cluster setting) feels a bit redundant with the analysis of a 2-point distribution. Finally, the experiments in Section 4 might be, I believe, too much focused on comparing RSGAN and JSGAN (in particular Table 2), while it is already known from [42: Jolicoeur-Martineau et al.] that RSGAN performs slightly better.

Correctness: I believe that the claims and the method are correct.

Clarity: The paper is relatively well-written. One remark would be that in Section 2, there may be too many claims, corollaries, definitions, propositions, and theorems (one right after the other): it can be a bit overwhelming.

Relation to Prior Work: I believe there could be more discussions - with previous works on local equilibriums in GANs. - with the objectives improving on the JSGAN to solve the mode collapse such as the DRAGAN proposition. In particular, this last paper "On convergence and stability of GANs" already hypothesized the existence of undesirable local equilibria.

Reproducibility: Yes

Additional Feedback:

[Author Response · NeurIPS 2020]

We thank the reviewers for their valuable feedback.

**R1**-*More analysis of global landscape on LSGAN, WGAN, etc:* 1) Note Remark 1 (L155): the results can be extended
to a class of separable GANs and relativistic GANs (R-GANs). More specifically, Thm. 3 (L1501) and Thm. 4
(L1513) show that separable GANs in Eq. (30) have bad basins while R-GANs in Eq. (31) have no bad basins. These
results only require minor conditions on the loss (Assumption J.1-J.5), covering logistic loss, hinge loss, squared
loss, etc.; 2) To cover LS-GAN (min-max version), two minor changes suffice: change the two $h$ functions to $h_1$
and $h_2$ in Eq. (30); change Assumption J.1-J.3 accordingly. We'll modify to include LS-GAN and R-LS-GAN. 3)
W-GAN is difficult to analyze. See App. J.2 for a discussion. The difficulty also indicates that our contribution goes
beyond a global landscape analysis in that we identify the losses (R-GANs) that are amenable to rigorous analysis.

**R1&R4**-*Experiments to validate proposed GAN losses.* As stated in L230, "the
effectiveness of relativistic GANs has been justified (to some extent)" and "our
goal is to use experiments to support the landscape theory." For this, we focus
on: a) **advantage in narrow nets**; b) **robustness to initialization**. Our paper
validates a) and b) in four ways: **1)** On L265-269 we show that for MNIST and a
certain initial point, RS-GAN outperforms JS-GAN by 30 FID scores (around 30
vs. 60). **2)** On L256, we show RS-GAN outperforms JS-GAN by 9 FID scores

Figure 1: LSUN ($256\times256$) generation with CNN structure for JS-GAN (above) and RS-GAN (bottom).

(45 vs. 53) when using a ResNet (bottleneck) on STL. **3)** In Tab. 11 (in appendix)
we show that R-hinge-GAN outperforms hinge-GAN with 1/4 width (24 vs. 33
FID on CIFAR10). Both SN-GAN and BigGAN papers use hinge-GAN, so we
check hinge loss. **4)** In experiments (Tab. 2, Tab. 11 in paper), separable versions (JS-GAN, hinge-GAN) do not beat
their relativistic counterparts (RS-GAN, R-hinge-GAN) in any case. These points show: R-GANs are more robust to
initialization and architecture. In new experiments we show: **5)** R-LS-GAN outperforms LS-GAN by 6 FID (42 vs. 48)
with 1/4 width (Tab. 1 below); **6)** RS-GAN outperforms WGAN-GP (Tab. 2 below); **7)** experiments on LSUN (higher
resolution than CIFAR10 - Fig. 1). These new experiments further justify the advantage of R-GANs. We will explain in
the main text.

**R2**-*existence of reachable path doesn't mean SGD can follow it; there are still questions.* There're two branches of re-
sults for neural nets: one branch [60,43] only discusses paths and basins assuming width $n$. A drawback, as you point
out: convergence of GD is not proved. Nevertheless, this is enough to distinguish RS and JS-GAN. Another branch
[2,25,37] proves convergence of SGD assuming width $\geq n^6$. Drawback: assumption of width $n^6$ is impractical. An
ideal result that SGD converges for width $n$ is a huge open question for neural nets (attempts exist, but all have strong
limitations). We do not intend to solve this open question here. We combine Thm. 1,2 with the first branch since it is
cleaner and already non-trivial. It is possible to combine with the second branch (on convergence), but it will make this
paper much longer. Future advances for neural nets can be potentially combined with our function space result.

**R2**-*parameterized result is vague. More detail of connections.* **and**
**R3**-*Intuition on proof of Prop. 1,2; how proof differs from supervised learning.* The proof strategy of Prop. 1,2 is an
adaptation of those of [60,43,59] to GANs. References [60,43,59] consider a convex loss (e.g. quadratic) in function
space and "transfer" decreasing paths in function space to decreasing paths in parameter space. To achieve this "transfer,"
some assumptions on the architecture (e.g. width large enough) are needed [60,43,59]. We apply this approach to the
GAN loss. In our proof, we state the general requirement of "transfer" in Assumption I.1-I.3, and then prove when these
assumptions hold in Appendix I.2 and I.3 (using architecture assumptions of [60,43,59]). We'll discuss in the main text.

**R3**-*proof sketch of Thm. 1, Thm. 2.* For Thm. 1, careful computation suffices. For Thm. 2, we build a graph with nodes
being input data, decompose the graph into cycles and trees, compute the loss by grouping the terms according to cycles
and trees, and add each term. We'll sketch the proof in the main text.

**R4**-*study WGAN.* Thanks for suggesting. i) See the first response, point 3) in L8 of the rebuttal. ii) We add simulation
showing that RS-GAN outperforms WGAN-GP for standard datasets (Tab. 2 below).

**R4**-*study DRAGAN architecture.* Thanks for pointing out the reference, which we read with great interest. It suggested
that mode collapse may be due to bad equilibria. However, there is no formal statement or proof. We will cite it and
discuss the connection with our work. DRAGAN adds a penalty of the gradient which may help eliminate some basins,
but it likely creates other basins. A formal analysis requires much effort, and is an interesting future direction.

**R4**-*Sec. 3 (two-cluster) a bit redundant with analysis of 2-point distribution.* Thanks. The analysis of a 2-point distri-
bution is about the landscape, not the training process. In contrast, Sec. 3 shows that the basin really appears in training,
and the theoretical values 0.48 and 0.35 really play a role in understanding the training process. Following your
comment, we will reduce the length of Sec. 3.

|  | Regular | channel/2 | channel/4 |
|---|---|---|---|
| LS-GAN | **32.93** | 37.83 | 48.63 |
| R-LS-GAN | 34.78 | **34.34** | **42.86** |

Table 1: FID results on CIFAR-10 for LS-GAN and R-LS-GAN with CNN structure given in Tab. 5 of the appendix.

|  |  | Regular | channel/2 | channel/4 |
|---|---|---|---|---|
| CNN | WGAN-GP | 39.66 | 42.39 | 50.56 |
|  | RS-GAN | **27.16** | **32.74** | **49.74** |
| ResNet | WGAN-GP | 21.33 | 23.80 | 40.45 |
|  | RS-GAN | **19.31** | **21.78** | **31.26** |

Table 2: FID results on CIFAR-10 for WGAN-GP and RS-GAN with CNN and ResNet.

[Meta-Review · NeurIPS 2020]

This paper analyzes the optimization landscape for GANs, and show that the landscape for vanilla GANs can have many bad local optima that correspond to mode collapse problems. On the other hand this can be solved using ideas similar to RS-GAN. Overall this is a solid theoretical contribution to the optimization of GANs.